# TRIMR: VERIFIER-BASED TRAINING-FREE THINKING TRIMMING FOR EFFICIENT TEST-TIME SCALING

**Weizhe Lin**[1*], **Xing Li**[2*], **Zhiyuan Yang**[2*], **Xiaojin Fu**[2*], **Hui-Ling Zhen**[2],
**Yaoyuan Wang**[1], **Xianzhi Yu**[2], **Wulong Liu**[2], **Xiaosong Li**[1], **Mingxuan Yuan**[2]
[1]Huawei Advanced Computing and Storage Lab  [2]Huawei Noah's Ark Lab
{linweizhe1, li.xing2, yangzhiyuan123, fuxiaojin,
 zhenhuiling2, wangyaoyuan1, yuxianzhi,
 liuwulong, lixiaosong20, yuan.mingxuan}@huawei.com

## ABSTRACT

Large Reasoning Models (LRMs) demonstrate exceptional capability in tackling complex mathematical and logical tasks by leveraging extended Chain-of-Thought (CoT) reasoning. Test-time scaling methods—such as prolonging CoT with explicit token-level exploration—can push LRMs' accuracy boundaries, but they incur significant decoding overhead. A key inefficiency source is LRMs often generate redundant thinking CoTs, which demonstrate clear structured overthinking and underthinking patterns. Inspired by human cognitive reasoning processes and numerical optimization theories, we propose *TrimR*, a **verifier-based, training-free, efficient framework to trim reasoning and enhance test-time scaling**, explicitly tailored for production-level deployment. Our method employs a lightweight, pretrained, instruction-tuned verifier to detect and truncate redundant intermediate thoughts of LRMs without any LRM or verifier fine-tuning. We present both the core algorithm and asynchronous online system engineered for high-throughput industrial applications. Empirical evaluations on **Ascend NPUs and vLLM** show that our framework delivers substantial gains in inference efficiency under large-batch workloads. In particular, on the four MATH500, AIME24/25, and GPQA benchmarks, **the reasoning runtime of QwQ-32B, DeepSeek-R1-Distill-Qwen-32B, and Pangu-R-38B is improved by up to 70% with negligible impact on accuracy**.

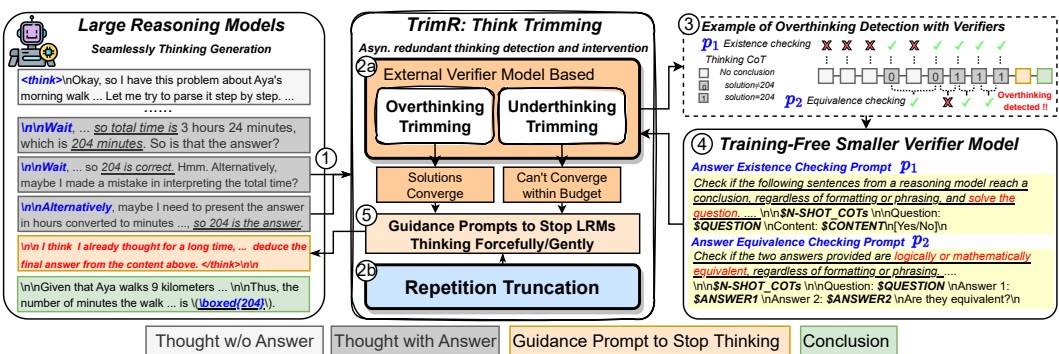

**Figure 1:** The TrimR framework. (1) CoTs are divided into sub-thoughts with reflection tokens in Sec. 3.1; (2a) Over/underthinking detection through answer convergence analysis in Sec. 3.2, 3.3 (2b) repetition truncation in Sec. 3.4; (3) By simplifying redundant detection as the binary answer existence and equivalence classification with prompts $p_1$ and $p_2$ in Appendix F, 7B instruction models replace PRMs/ORMs, avoiding instability from full-sequence scoring; (4) The verifier applies $p_1$ to assess conclusion completeness in individual thoughts and $p_2$ to identify overthinking when consecutive thoughts yield identical answers. Early termination is triggered with Algorithm 1, 2; (5) Thinking termination prompts for LRMs are generated based on verifier decisions to halt redundant reasoning gently or forcefully in Sec. 3.5.

---

*Equal contribution.

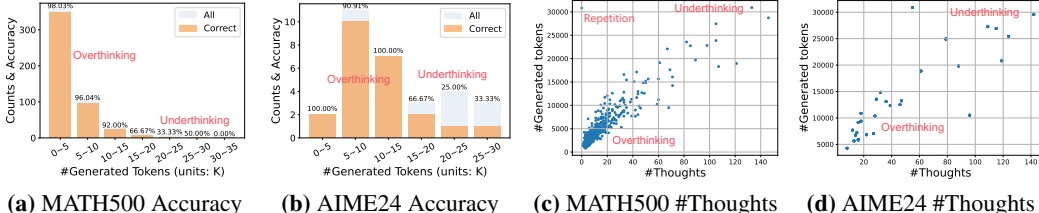

| **(a)** MATH500 Accuracy | **(b)** AIME24 Accuracy | **(c)** MATH500 #Thoughts | **(d)** AIME24 #Thoughts |

**Figure 2:** Correlation between occurrence, accuracy, and the number of generated thoughts/tokens of QwQ-32B on MATH500 (Hendrycks et al., 2021) and AIME24 (AIME2024). For hard questions, QwQ can spin out up to 140 lengthy yet incorrect thoughts—indicating a need for underthinking trimming. For easier questions, QwQ delivers brief, highly accurate answers—yet there's still room to make them even more concise by overthinking trimming.

# 1 INTRODUCTION

Large Reasoning Models (LRMs) such as OpenAI o1 (OpenAI, 2024), DeepSeek R1 (Guo et al., 2025), and Qwen QwQ (Team, 2025) achieve expert-level performance on *mathematical and scientific reasoning tasks* by decomposing problems into manageable subproblems, conducting step-by-step analysis, detecting and correcting errors, and exploring alternative solutions. However, this extended "thinking" incurs substantial decoding overhead and inference cost, hindering deployment in resource-constrained settings (Guo et al., 2025).

Benchmarking on AIME24 (AIME2024) and MATH500 (Hendrycks et al., 2021) reveals two key inefficiencies: **overthinking** and **underthinking** (Marjanović et al., 2025). **Overthinking** manifests as redundant verification of already-correct steps—often signaled by reflection tokens such as "Wait" or "Alternatively"—which increases output length without accuracy gains (as in Fig. 1, 2). **Underthinking** occurs on harder problems when the model oscillates among incomplete reasoning chains and fails to converge, producing lengthy yet inaccurate responses (Fig. 2). Representative examples are provided in Appendix G.

The extended Chain-of-Thought (CoT) reasoning in LRMs poses deployment challenges: decoding lengths vary widely and runtime scales superlinearly with sequence length. Training-based methods such as (Yan et al., 2025; Chen et al., 2024; Munkhbat et al., 2025; Yu et al., 2024) reduce token generation in LRMs but impose heavy training and computational costs on large models and may degrade their general capabilities. In contrast, training-free techniques integrate seamlessly and preserve original behavior: TokenBudget (Han et al., 2024) dynamically adjusts token budgets via prompting, and Chain of Draft (Xu et al., 2025) uses concise instructions to shorten output, yet both require invasive inference-time modifications.

We thus propose *TrimR*, a **verifier-based, training-free approach for online thinking trimming** while maintaining reasoning performance. We detect overthinking and underthinking in **intermediate thought answers** with a **lightweight verifier and trigger prompt-based LRM thinking early termination upon redundancy detection**. By simply checking answer existence and equivalence in brief thought segments, our method leverages compact verifier models instead of complex Process Reward Models (PRMs) or Outcome Reward Models (ORMs) (Lightman et al., 2024). The non-invasive early termination preserves original LRM capabilities. Finally, we present an online asynchronous system collaborating LRMs and small verifiers to support industrial-scale workloads. **TrimR offers a significant advantage by maintaining the performance and knowledge of LRMs** through a targeted intervention strategy that is activated only when redundant reasoning is detected.

**Notations.** Given an input $X$, the LRM $\Pi$ generates a response $Y = \Pi(X)$. We denote $\mathbf{y}_{<t} = [y_1, y_2, ..., y_{t-1}]$ as the previously generated tokens. Each time $y_t = p_\Pi(X, \mathbf{y}_{<t})$. With a slight abuse of notation, let $\Pi(X, \mathbf{y}_{<t})$ denote full response including $\mathbf{y}_{<t}$.

**Theoretical Foundation.** Our method unifies human cognitive heuristics—overthinking and underthinking as confidence-threshold and diminishing-returns processes—with mathematical optimization. People typically stop thinking further after finding answers to simple questions and give up on complex tasks after too many unsuccessful attempts. We model reasoning process as an optimization problem in "language space", where LRMs traverse token trajectories and converge to an optimal

solution. Mirroring numerical optimizers' early-stopping, we introduce a termination criterion that halts reasoning once reasoning converges or marginal gains fall below a preset threshold. Formally, given an input $X$ with a partial response $\mathbf{y}_{<t}$, the reasoning performance of LRM $\Pi$ is denoted as $Perf(X, \mathbf{y}_{<t}|\Pi)$. We derive a compression rule $\mathbf{c}$ to determine the stopping time $t' \in \{0, ..., t-1\}$. The goal is to minimize the inference cost of $\mathbf{y}_{<t'}$ referred to as $Infer\_Cost(\mathbf{y}_{<t'})$ without degrading reasoning performance.

$$\min_{\mathbf{c}(\cdot)} \quad Infer\_Cost(\mathbf{y}_{<t'}) \quad \text{s.t.} \quad t' = \mathbf{c}(\mathbf{y}_{<t})$$

$$Perf(X, \mathbf{y}_{<t'}|\Pi) \geq Perf(X, \mathbf{y}_{<t}|\Pi)$$

In designing TrimR, we conceptualized LRM reasoning as an iterative optimization-style search, combining hypothesis generation, verification, and reflective exploration to avoid local optima. Similar to classical optimization practice, reasoning can be terminated when multiple explorations converge to consistent outcomes or the model fails to break a performance plateau. Overly permissive stopping criteria risk premature termination. Based on this insight, TrimR captures a verifier-supported stopping condition that achieves resource reduction while preserving accuracy. In this context, $Perf(X, y)$ denotes the performance of a generated reasoning sequence $y$ for problem $X$ (e.g., benchmark accuracy), and the compression rule $\mathbf{c}$ (TrimR) aims to identify the shortest prefix $y_{<t}$ such that reasoning can safely terminate.

**Contributions.**

- We propose a lightweight, training-free method for dynamic thinking trimming in LRMs—mitigating both over- and underthinking—via a small verifier, enabling efficient test-time scaling with negligible loss of reasoning accuracy.
- We develop an asynchronous thinking trimming system for industrial-scale deployment, seamlessly integrating with existing inference infrastructures.
- Through extensive evaluations and ablations on diverse models (QwQ-32B, Deepseek-R1-Distill-Qwen-32B, and Pangu-R-38B), we demonstrate consistent reductions in reasoning cost and token usage across standard reasoning benchmarks in production settings.

## 2 RELATED WORKS

This section reviews recent advancements in efficient reasoning techniques for LRMs. For a more detailed discussion, see comprehensive surveys (Sui et al., 2025; Qu et al., 2025; Su et al., 2025). The slight difference of the fast and slow thinking models motivates efficient long-to-short reasoning with model merging, which directly fuse weights of models with different thinking patterns using carefully designed algorithms (Team et al., 2025; Wu et al., 2025a). Apart from it, there are three types of commonly used efficient reasoning methods.

**Training-based Methods.** Approaches in this category modify or fine-tune LRMs to generate concise reasoning. Reinforcement learning with length-based rewards prunes verbose chains (Hou et al., 2025; Xia et al., 2025; Zhang et al., 2025a; Team et al., 2025; Chen et al., 2024; Arora & Zanette, 2025; Chen et al., 2025; Fang et al., 2025), while supervised fine-tuning teaches models to compress explanations (Munkhbat et al., 2025; Yu et al., 2024). Latent-space techniques further minimize token usage by operating in a compact semantic embedding space (Hao et al., 2024). Although effective, these methods demand substantial compute, risk task-specific overfitting, and may degrade general-purpose capabilities.

**Self-Evaluation Methods.** These techniques prompt LRMs to assess their own confidence and decide when to stop reasoning. Adaptive schemes ask the model to predict the benefit of restarting (Manvi et al., 2024) or to estimate certainty at key junctures (Yang et al., 2025). SelfThink (Zeyu et al., 2025) is proposed to facilitate LRMs' intrinsic task complexity classification capabilities to dynamically switch between fast and slow thinking. ST-BoN leverages embedding-distance metrics for early truncation (Wang et al., 2025b). While they avoid external models, the added inference steps may introduce latency. Some contemporaneous works intervene in inference to reduce token usage in reasoning: SpeedAdapt (Lin et al., 2025) adjusts hidden states dynamically to control the reasoning speed; NoThink (Ma et al., 2025) suppresses reasoning entirely; AlphaOne (Zhang et al., 2025b) modulates slow and fast thinking at test time based on average thinking length.

**Model Collaboration.** Hybrid frameworks use auxiliary evaluation models, reward models, or thought proposers to guide decoding. Dynasor monitors semantic entropy and reward-model outputs

for early stopping (Fu et al., 2024; Kuhn et al., 2023). Speculative Rejection uses partial-output scores from a reward model to terminate best-of-$N$ search (Sun et al., 2024). However, above works heavily rely on the performance of the reward model. Another work (Xi et al., 2024) trains a critique model which provides step-level feedback to guide the reasoning model on the fly. There are also concurrent works on improve LRM efficiency by generating thoughts with smaller reasoning models for speculative reasoning (Pan et al., 2025; Wang et al., 2025a). CoThink (Fan et al., 2025) leverages an instruction model to guide reasoning and reduce reasoning steps.

Unlike prior work, our TrimR framework is training-free and non-invasive: a lightweight verifier dynamically detects and helps truncate redundant reasoning. No extra self-evaluation steps are introduced, so our method can be easily integrated into existing inference frameworks.

## 3 METHOD

We propose an **efficient, verifier-based, training-free thinking trimming** algorithm that dynamically prunes redundant CoT generation during online inference. Our algorithm replicates human cognitive mechanisms which utilize internal verifiers to check and stop thinking. We introduce smaller verifiers to detect redundant thinking without fine-tuning verifiers or LRMs. Designed for industrial-scale batch processing, our framework (Fig. 1) comprises three modules: **overthinking trimming** (Sec. 3.2), **underthinking trimming** (Sec. 3.3), and **repetition truncation** (Sec. 3.4). Guidance prompts for halting reasoning are detailed in Sec. 3.5. The online system is presented in Sec. 3.6.

### 3.1 THOUGHT DETECTION AND ANSWER EXTRACTION

The reasoning thoughts have clear structured patterns and are usually separated with **reflection tokens**, such as "$\backslash n \backslash nWait$", "$\backslash n \backslash nBut$", and "$\backslash n \backslash nAlternatively$". In addition, LRMs normally generate answers at the end of thoughts and then verify them as in Fig. 1.

During thinking, we periodically segment the thinking process into sub-thoughts $[r_1, r_2, ..., r_k]$:

$$Think\_Seg(\mathbf{y}_{<t}) = [r_1, r_2, ..., r_k].$$

Here, we separate it by reflection tokens (full list in Appendix E). When new reflection tokens are detected asynchronously, we split the whole thought between two consecutive reflection tokens into sentences and concatenate the last several sentences of the current thought as the extracted **intermediate thought answers**. Formally, The last $N_{sent}$ sentences of each sub-thought $r_l$ are denoted as $s_l$, i.e. $s_l = Last\_sentences(r_l, N_{sent})$. The last several sentences are shorter but more informative than the whole sub-thought and normally contain answers as demonstrated in Fig. 1. We do not further process the answer but only skip the extremely short thoughts, which normally do not contain useful thoughts.

We detect predefined reflection tokens asynchronously so that LRMs continue generating tokens without throughput loss. Since reflection tokens account for only 0.5% of all outputs and thoughts are significantly less than tokens as in Fig. 2, the verifier is invoked infrequently and can serve multiple LRMs concurrently as analyzed in Appendix M. It makes the overhead of redundancy checks negligible compared to the gains from thinking compression.

### 3.2 OVERTHINKING TRIMMING

In overthinking scenarios, LRMs typically arrive at correct solutions using only $30 \sim 50\%$ of the total generated tokens, yet continue producing redundant reasoning paths or alternative justifications before finalizing an answer (Chen et al., 2024). While this may aid complex or uncertain tasks, it burdens simpler ones with uninformative content and higher latency and inference costs without accuracy gains. We propose an overthinking detection and compression algorithm that uses lightweight verifier models to prune redundant reasoning while preserving LRM accuracy. It emulates verifier confidence based human thinking termination and convergence-based early termination mechanisms of numerical optimizers.

**Simple Tasks to Utilize Smaller Models.** We simplify overthinking detection as answer existing and equivalence checking, which are simpler binary classification problems than scoring the whole

sequences. By reducing detection to checking answer existence and comparing final outputs of consecutive intermediate thought answers, we can deploy compact (7B) verifiers with satisfactory language comprehension and instruction following capabilities without fine-tuning, greatly lowering inference overhead compared to full-sequence reward models (Lightman et al., 2024; Liu et al., 2025). Moreover, training PRMs/ORMs for accurate scoring is complex and often yields unstable rewards on identical sequences, undermining reliable overthinking detection (Sun et al., 2024).

**Two-Stage Verification.** To minimize intervention from answerless intermediate checks, we introduce two-stage verification. First, we confirm the candidate solution is present in the current reasoning thought. We then verify that intermediate thought answers are semantically or logically equivalent, regardless of correctness. Both tasks can be handled by the lightweight verifier $\mathcal{F}_v$: x

*(1) Answer existence checking*: The set of thoughts with solutions $S^*$ is defined as:

$$S^* = \{ s_i \mid \mathcal{F}_v(p_1(s_i)) \};$$

$$\mathcal{F}_v(p_1(s_i)) = \mathbb{I}\Big[ p_{\mathcal{F}_v}(y = \text{"Yes"} \mid p_1(s_i)) $$
$$> p_{\mathcal{F}_v}(y = \text{"No"} \mid p_1(s_i)) \Big]. \tag{1}$$

where $p_1(s_i) = p_1(X, s_i)$ ($X$ omitted for simplicity) is the verifier prompt for answer existence taking the problem $X$ and thought answer $s_i$ as parameters.

*(2) Equivalence checking*: The equivalence $r$ between two consecutive thoughts in $S^*$ is computed as:

$$r(s_i^*, s_{i+1}^*) = \mathcal{F}_v\big(p_2(s_i^*, s_{i+1}^*)\big)$$
$$= \mathbb{I}\Big[ p_{\mathcal{F}_v}(y = \text{"Yes"} \mid p_2(s_i^*, s_{i+1}^*))$$
$$> p_{\mathcal{F}_v}(y = \text{"No"} \mid p_2(s_i^*, s_{i+1}^*)) \Big]. \tag{2}$$

where $p_2(s_i, s_j) = p_2(X, s_i, s_j)$ ($X$ omitted for simplicity) is the verifier prompt for answer equivalence, taking the problem $X$ and two consecutive thought answers in $S^*$ that both achieve solutions.

---

**Algorithm 1:** Overthinking Compression

**Input:** Input $X$, repeat-threshold $M$
**Output:** Generated output $Y$
$stopped \leftarrow False;$   $y_t \leftarrow p_\Pi(X);$
$prev\_concluded\_thought \leftarrow None;$   $count \leftarrow 0;$
**while** $y_t \neq <eos>$ **do** // LRM iteratively generates $y_t$
  // Periodically segment and check for a new concluded thought
  // $\mathcal{F}_v(p_1(s_l))$ defined in Eqn. 1, $r(s_i, s_j)$ defined in Eqn. 2.
  **if** $y_t = </think>$ **then**
   |  $stopped = True, break;$
  **end**
  **if** *a new segment $s_\ell$ is found and $\mathcal{F}_v(p_1(s_l)) = 1$ and stopped $= False$* **then**
   | **if** $prev\_concluded\_thought \neq None$ **then**
   |  | **if** $r(prev\_concluded\_thought, s_\ell) = 1$ **then**
   |  |  |  $count \leftarrow count + 1;$
   |  | **else**
   |  |  |  // reset on any mismatch
   |  |  |  $count \leftarrow 0;$
   |  | **end**
   | **end**
   | **if** $count \geq M$ **then**
   |  | // early-stop after $M$ repeats
   |  | $y_t \leftarrow p_\Pi((X, \mathbf{y}_{<t}, stop\_tokens));$
   |  | $stopped \leftarrow True, break;$
   | **end**
   | $prev\_concluded\_thought \leftarrow s_\ell;$
  **else**
   |  $y_t \leftarrow p_\Pi((X, \mathbf{y}_{<t}, y_t));$
  **end**
**end**
**return** $(\mathbf{y}_{<t}, y_t)$

---

**Verifier Prompts.** Verifiers take the answer existence prompt $p_1$ and answer equivalence prompt $p_2$ and directly return the binary classification results with the probabilities of "Yes" and "No" tokens. Therefore, only prefilling of verifiers is required, which reduces the computational cost. The two prompts for verifiers ($p_1$ and $p_2$) in Fig. 1 with placeholders are available here. Details of placeholders are in Appendix F. The answer existence checking prompt $p_1$ consists of system prompt for ignoring unimportant formats and phrases, n-shot CoTs with positive and negative examples, the current question, and the intermediate answer. Similarly, the verifier only checks if two consecutive intermediate thought answers are semantically or logically equivalent with the prompt $p_2$.

---

**Verifier Prompt $p_1$ for Answer Existence:**
Check if the following sentences from a reasoning model reach an answer, regardless of formatting or phrasing, and solve the question. Return 'Yes' if the content finds a solution, otherwise 'No'. Return only 'Yes' or 'No' with no explanation.
*$N-SHOT\_COTs (details in Appendix)*
Question: *$QUESTION*
**Content:** *$CONTENT* \n  [Yes/No]:

---

**Verifier Prompt $p_2$ for Answer Equivalence**
Check if the two answers provided are logically or mathematically equivalent, regardless of formatting or phrasing. Return 'Yes' if they are equal in meaning/value and valid solutions to the question, otherwise 'No'. Return only 'Yes' or 'No' with no explanation.
*$N-SHOT\_COTs (details in Appendix)*
Question: *$QUESTION*
Answer 1: *$ANSWER1*
Answer 2: *$ANSWER2*
Are they equivalent? [Yes/No]:

---

**Early Termination.** We implement early termination through the prompt mechanism introduced in Sec. 3.5, specifically when the model consecutively agrees with previous reasoning steps $M$ times. The process is described in Algorithm 1. We only consider thoughts in $S^*$ to bypass intermediate thoughts lacking definitive solutions. Such a protocol analogously replicates human cognitive patterns where reasoning halts upon achieving $M + 1$ consecutive consistent solutions, paralleling the convergence termination criterion in optimization algorithms.

**Resource Saving.** By limiting inputs to 200–400 tokens instead of processing full reasoner outputs (8K–128K tokens) as did in PRMs/ORMs (Sun et al., 2024), we drastically cut memory and compute overhead, boosting batch throughput and reducing verification latency with **shorter verifier prompts**. Also, we always reuse the KV cache for the system prompt and question with **prefix caching**, then batch answers in triplets—remapping the second answer to a fixed placeholder ($ANSWER1 in $p_2$). This lets us retain its cache across examples, so only the cache for the other placeholder ($ANSWER2, used by the first and third answers) needs updating.

## 3.3 UNDERTHINKING TRIMMING

In particularly difficult tasks, LRMs repeatedly verify intermediate steps—an indicator of uncertainty that leads to divergent, redundant reasoning. We find that when the model oscillates between different thought paths, it seldom converges: proposed solutions often fail verification, triggering further exploration that inflates latency (Fig. 6) without boosting accuracy, and excessive diversity can even hinder convergence (Chen et al., 2024).

To mitigate this, we repurpose the verifier's answer-existence/consistency feedback to detect hesitant overthinking. Concretely, if after $R_{thres}\%$ of the token budget and $N_{thres}$ reasoning rounds, the model has not produced at least three consecutive answer-bearing chains that agree on the same solution, we flag the run as underthinking (Appendix B Algorithm 2). At that point, a guidance prompt instructs the model to halt further reasoning and summarize its existing insights. It mimics human tendency to abandon difficult tasks that exceed its capabilities after multiple failed attempts.

This mechanism relies on two principles: (1) repeated convergence on the same solution indicates that the reasoning process is coherent and sufficiently thorough; and (2) the thresholds $R_{thres}$ and $N_{thres}$ must be calibrated to afford the model adequate opportunity to explore alternative solution paths before termination. Ablation studies on these thresholds are presented in Appendix J.

## 3.4 REPETITION TRUNCATION

Recurrent token loops in LRMs often persist despite probabilistic sampling tweaks (e.g., high temperature), especially when sequence length increases. Robust repetition detection and truncation is therefore essential to prevent wasted computation and improve user experience. Our solution uses a rolling-hash–based detector to identify repeating token-ID subsequences in real time. By updating hashes incrementally for each new token, we avoid recomputing whole-sequence hashes, enabling efficient, on-the-fly repetition checks. This dynamic algorithm is enabled by default unless otherwise specified. The effectiveness of this module is provided in Appendix I.

## 3.5 GUIDANCE PROMPTS FOR LRMS THINKING COMPRESSION

We devised two **gentle and forceful guidance prompts** (presented in Appendix F) to guide terminating the reasoner's thought process. The gentle prompt curbs overthinking by steering the model toward concluding its reasoning with `**Final Answer**\n`, whereas the forceful prompt both prevents underthinking and breaks any repetitive loops that occur before the designated "think end" token (e.g. `</think>`) is emitted. In our experiments, the forceful prompt consistently halted endless generation during the reasoning stage.

## 3.6 ONLINE SYSTEM DESIGN

Our inference system is built on vLLM (Kwon et al., 2023) (v0.6.0) running on Ascend-910B-64GB NPUs, though it can be integrated equally well with other inference frameworks (e.g. SGLang (Zheng et al., 2024)) or hardware platforms (e.g. GPUs). As shown in Fig. 3, it comprises two tightly coupled elements: the inference engine itself, which orchestrates decoding via vLLM, and the Test-Time Thinking Trimming System (T4S), which performs real-time reasoning compression in parallel. T4S is structured around a **Reasoning Verifier**—responsible for detok-

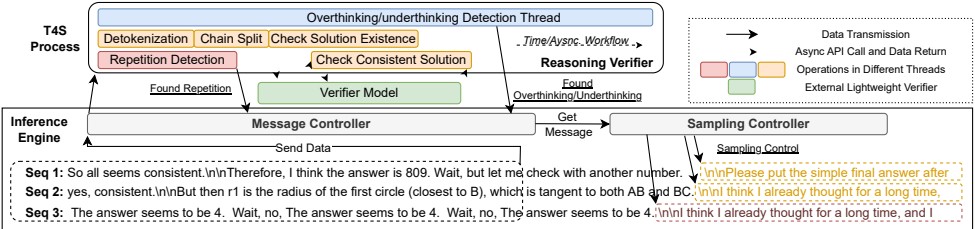

**Figure 3:** System design for the Test-Time Thinking Trimming System (T4S). The figure shows three sequences that are flagged as (1) overthinking; (2) underthinking; and (3) repetitive generation. The inference engine streams updates through the Message Controller into the external T4S process, which issues asynchronous API calls to a lightweight verifier.

enizing output into discrete reasoning chains, detecting overthinking or underthinking, and using an external small verifier model to confirm solution validity—a **Message Controller** that exchanges operational data (prompt and generated token IDs) with the verifier at configurable intervals (per $N_{send}$ tokens generated), and a **Sampling Controller** that adaptively modifies output logits to enforce specific token generations.

## 4 EXPERIMENTS

### 4.1 EXPERIMENT SETUP

**Benchmarks.** This work focuses on mathematical and scientific problems: *Mathematical benchmarks:* **MATH500** (Hendrycks et al., 2021), **AIME24** (AIME2024), **AIME25** (AIME2025) and *Scientific benchmarks:* **GPQA Diamond** (Rein et al., 2024). Details are available in Appendix H. We additionally include results on LiveCodeBench (Jain et al., 2024) for dense and MoE models. As the primary emphasis of this work is on mathematical and scientific reasoning, code tasks are included for supplementary reference in Appendix O.

**Metrics.** Apart from accuracy (**Acc.**), we mainly care about the efficiency metrics. **Runtime** denotes the total wall-clock time to process all requests in each dataset. **TPR** is the average Time Per Request, while **TPR-T90** is the TPR of the fastest/top 90% requests. **#Tokens(M)** is the number of generated tokens in millions. Runtime represents the total waiting time of requests, while lower TPR and TPR-T90 indicate better single user experience and higher Queries-per-Second (QPS).

**Configurations.** All the experiments are conducted on servers with 8 Ascend 910B-64GB NPUs and 192-cores Kunpeng-920 CPUs with 1.5TB memory. All dataset requests (e.g., the 500 questions in MATH500)[1] are submitted to vLLM concurrently, and we record key metrics such as total wall-clock time and per-request latency. We benchmark two open-source models (QwQ-32B and DeepSeek-R1-Distill-Qwen-32B(R1Q-32B)) alongside the closed-source Pangu-R-38B, using fixed input/output lengths of 2K/30K tokens. *For readers' interest, the results on Pangu Pro MoE (Tang et al., 2025), publicly released after the date of this study, are presented in Appendix N.* Although extending outputs to ~128K tokens yields marginal gains, such settings are impractical for production, so we cap the output at 30K. $M$=2, $N_{send}$=50. The verifier is Pangu-7B (open-sourced) (Pangu Team, 2025) (by default) and Qwen2.5-7B-Instruct (Qwen Team, 2024) (Table 4).

### 4.2 MAIN RESULTS

Table 1 shows that introducing TrimR delivers consistent and substantial efficiency gains across all three models and four benchmarks (up to 70% runtime reduction), with accuracy largely unaffected (less than 1.7% drop). Specifically, **runtime** is reduced by 16–39% for QwQ-32B (e.g., from 4,413s to 3,118s on MATH500, –29.3%), 19–39% for Pangu-R-38B(–33.8% on MATH500), and an impressive 53–70% for R1Q-32B (–67.0% on MATH500; –70.0% on GPQA Diamond). Similar reductions are seen in **TPR** (e.g., R1Q-32B's TPR on AIME24 drops from 3,717.6s to 1,433.9s, –61.4%). **Token usage** also drops by 8–46% overall, with R1Q-32B showing the largest reduction (from 2.447M to 1.320M tokens, –46.1% on GPQA Diamond). Despite these gains, **accuracy** is preserved or even improved. QwQ-32B gains on MATH500 (+1.2%) and AIME25 (+0.8%), while R1Q-32B improves 2.0–13.2% on three benchmarks. *For readers' interest, we discussed the reasons*

---

[1]Since AIME has only 30 questions, we replicate it eightfold to ensure the engine receives enough requests.

**Table 1:** Performance comparison of QwQ-32B, Pangu-R-38B, and DeepSeek-R1-Distill-Qwen-32B on the MATH500, AIME24, AIME25, and GPQA Diamond benchmarks. Relative improvements are highlighted in green, and regressions in red.

| Model | Runtime(s) | TPR(s) | TPR-T90(s) | Acc. | #Tokens(M) |
|---|---|---|---|---|---|
| **MATH500** | | | | | |
| QwQ 32B | 4413 | 593.1 | 439.7 | 95.6% | 2.278 |
| *w/ TrimR* | 3118 -29.3% | 499.7 -15.7% | 377.4 -14.2% | 96.8% 1.2% | 1.953 -14.3% |
| DeepSeek-R1-Distill-Qwen-32B | 7602 | 733.1 | 278.8 | 92.4% | 2.219 |
| *w/ TrimR* | 2511 -67.0% | 315.8 -56.9% | 218.0 -21.8% | 94.4% 2.0% | 1.330 -40.1% |
| Pangu-R-38B | 3665 | 447.4 | 300.8 | 95.6% | 1.912 |
| *w/ TrimR* | 2426 -33.8% | 367.6 -17.8% | 264.6 -12.0% | 94.4% -1.2% | 1.551 -18.9% |
| **AIME24** | | | | | |
| QwQ 32B | 6992 | 2437.6 | 2138.6 | 76.6% | 3.189 |
| *w/ TrimR* | 4255 -39.1% | 1572.6 -35.5% | 1431.6 -33.1% | 76.6% -0% | 2.444 -23.3% |
| DeepSeek-R1-Distill-Qwen-32B | 10299 | 3717.6 | 3156.0 | 66.6% | 3.252 |
| *w/ TrimR* | 4799 -53.4% | 1433.9 -61.4% | 1228.7 -61.1% | 70.0% 3.3% | 2.096 -35.6% |
| Pangu-R-38B | 6164 | 1912.3 | 1639.4 | 78.3% | 2.466 |
| *w/ TrimR* | 3848 -37.6% | 1299.4 -32.0% | 1154.8 -29.6% | 76.6% -1.7% | 2.006 -18.6% |
| **AIME25** | | | | | |
| QwQ 32B | 7436 | 2771.5 | 2513.8 | 60.0% | 3.426 |
| *w/ TrimR* | 6215 -16.4% | 2302.5 -16.9% | 2032.4 -19.2% | 60.8% 0.8% | 3.070 -10.4% |
| DeepSeek-R1-Distill-Qwen-32B | 13055 | 5474.3 | 4861.7 | 47.9% | 3.932 |
| *w/ TrimR* | 6169 -52.7% | 1897.0 -65.3% | 1549.6 -68.1% | 56.3% 8.4% | 2.434 -38.1% |
| Pangu-R-38B | 9216 | 3053.4 | 2723.6 | 57.5% | 3.117 |
| *w/ TrimR* | 5591 -39.3% | 1958.6 -35.9% | 1731.6 -36.4% | 57.5% 0.0% | 2.470 -20.8% |
| **GPQA Diamond** | | | | | |
| QwQ 32B | 4406 | 1302.6 | 1115.0 | 66.0% | 1.572 |
| *w/ TrimR* | 3198 -27.4% | 1170.7 -10.1% | 1025.5 -8.0% | 65.2% -0.8% | 1.438 -8.5% |
| DeepSeek-R1-Distill-Qwen-32B | 11366 | 3568.0 | 2786.2 | 45.4% | 2.447 |
| *w/ TrimR* | 3411 -70.0% | 902.0 -74.7% | 720.4 -74.1% | 58.6% 13.2% | 1.320 -46.1% |
| Pangu-R-38B | 3120 | 994.7 | 866.8 | 59.1% | 1.378 |
| *w/ TrimR* | 2516 -19.4% | 901.2 -9.4% | 788.9 -9.0% | 60.1% 1.0% | 1.273 -7.6% |

*behind such improvements in Appendix K.* Minor regressions are all under 2%, a reasonable tradeoff for significant runtime reductions.

In Table 2, we include concurrent baselines (Certaindex (Fu et al., 2024), CoThink (Fan et al., 2025), SpeedAdapt (Lin et al., 2025); preprints at the time of submission, with some released after our initial draft). we find that TrimR consistently delivers substantial token savings while maintaining accuracy across datasets. For example, on R1Q-32B with AIME24, TrimR reduces token usage by 35.6% with stable accuracy, whereas CoThink achieves only a 12.5% reduction but suffers a sharp 13.3% accuracy drop. SpeedAdapt, which adjusts hidden states to control reasoning speed, maintains accuracy reasonably well but yields only 4.0-9.3% token savings on QwQ-32B.

**Table 2:** Comparing TrimR with baseline methods. Acc.: accuracy change relative to respective baseline; #Tok.: token usage reduction.

| Model | MATH500 | | AIME24 | |
|---|---|---|---|---|
| | Acc. | #Tok. | Acc. | #Tok. |
| **QwQ-32B** | | | | |
| CoThink | -3.0% | **-19.1%** | 3.3% | -16.2% |
| SpeedAdapt | 0.0% | -4.0% | 1.1% | -9.3% |
| TrimR (ours) | 1.2% | -14.3% | 0.0% | **-23.3%** |
| **DeepSeek-R1-Distill-Qwen-32B** | | | | |
| Certaindex | -4.0% | -19.0% | -4.0% | -15.0% |
| CoThink | -2.0% | -36.6% | -13.3% | -12.5% |
| SpeedAdapt | 0.7% | -7.3% | 1.5% | -12.7% |
| TrimR (ours) | 2.4% | **-40.1%** | 3.3% | **-35.6%** |

Certaindex introduces extra inference cost yet still underperforms TrimR in token reduction. Overall, TrimR strikes a more favorable balance between efficiency and reliability relative to the baseline methods.

## 4.3 ANALYSIS

**Effects of Trimming Methods.** Ablation results in Table 3 demonstrate that combining overthinking and underthinking trimming achieves the greatest efficiency gains with minimal accuracy trade-offs. For QwQ-32B on MATH500, overthinking trimming alone reduces TPR by 12.0% and tokens by 10.6% while improving accuracy by 1.2%, whereas underthinking trimming yields smaller gains (TPR: -4.8%, tokens: -3.3%) with a minor 0.6% accuracy drop. Their combination maintains the 96.8% accuracy while achieving TPR and token reductions of 15.8% and 14.3%, respectively.

For Pangu-R-38B, the combined approach reduces TPR by 17.8% and tokens by 18.9% on MATH500 with negligible 1.2% accuracy loss. R1Q-32B shows even stronger gains: combined

**Table 3:** Ablation analysis of overthinking and underthinking trimming, showing that both methods markedly reduce TPR and token usage without compromising reasoning accuracy.

| | MATH500 | | | AIME24 | | |
|---|---|---|---|---|---|---|
| | TPR | #Tokens | Accuracy | TPR | #Tokens | Accuracy |
| Pangu-R-38B | - | - | 95.6% | - | - | 78.3% |
| *w/ overthinking trimming* | -16.6% | -16.1% | 95.4% | -30.2% | -15.7% | 76.6% |
| *w/ underthinking trimming* | -10.5% | -8.3% | 95.8% | -26.0% | -13.7% | 75.4% |
| *w/ both* | -17.8% | -18.9% | 94.4% | -32.1% | -18.6% | 76.6% |
| QwQ 32B | - | - | 95.6% | - | - | 76.6% |
| *w/ overthinking trimming* | -12.0% | -10.6% | 96.8% | -13.1% | -12.4% | 76.6% |
| *w/ underthinking trimming* | -4.8% | -3.3% | 95.0% | -27.0% | -16.1% | 76.3% |
| *w/ both* | -15.8% | -14.3% | 96.8% | -35.5% | -23.3% | 76.6% |
| DeepSeek-R1-Distill-Qwen-32B | - | - | 90.4% | - | - | 60.0% |
| *w/ overthinking trimming* | -43.5% | -28.1% | 91.6% | -49.0% | -22.3% | 63.3% |
| *w/ underthinking trimming* | -41.4% | -25.0% | 92.8% | -57.1% | -29.8% | 63.8% |
| *w/ both* | -56.9% | -40.1% | 92.4% | -61.4% | -35.6% | 63.3% |

trim slashes TPR by 56.9% and tokens by 40.1% while boosting accuracy from 90.4% to 92.4%. Model-specific patterns emerge: QwQ-32B exhibits lower redundancy (overthinking TPR reduction: 12.0% vs. 43.5% for R1Q-32B). Conversely, R1Q-32B's high token usage (2.447 vs. 1.572 for QwQ-32B on GPQA) reflects frequent self-verification, which dynamic trimming mitigates. Variations in trimming efficacy across models and benchmarks (e.g., QwQ-32B: MATH500 TPR -12.0% vs. AIME24 -13.1%) underscore the need to apply both strategies to optimize efficiency across tasks.

**Effects on Distribution.** Fig. 4 depicts the empirical token-count distributions for our reasoning tasks before and after applying TrimR to QwQ-32B. In the original (untrimmed) setting, approximately 64% of problem instances fell within the lowest bin (0–5K tokens), with the remainder spread across higher token ranges (5–32K tokens). After trimming, this proportion rises to nearly 70%, and the frequency of "long-context" instances ($\geq$10K tokens) drops by over 25%. In particular, the heaviest tail (20–32K tokens) is reduced by more than two-thirds, from roughly 6% of cases down to under 2%. This pronounced leftward shift in the distribution demonstrates that TrimR effectively prunes superfluous context, lowering the average token footprint per query.

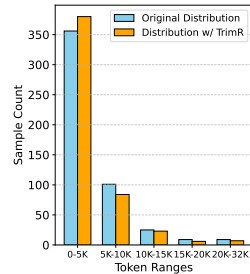

**Figure 4:** Token distributions with/without TrimR.

**Verifier Accuracy.** To assess chain-level consistency, we used Pangu-R-38B to generate full reasoning traces for all MATH500 questions, split them into T4S-defined chains, and manually annotated answer consistency for 684 randomly sampled adjacent chain pairs.

**Table 4:** Verification accuracy, and downstream performance of Pangu-7B and Qwen2.5-7B-Instruct verifiers on the MATH500 dataset. *Verifier Acc.(%)* denotes the fraction of correctly judged chain pairs; The rest are downstream performance on MATH500: *MATH500 Acc.(%)* is downstream task accuracy.

| Verifier | Verifier Acc.(%) | Runtime | TPR | MATH500 Acc.(%) | #Tokens(M) |
|---|---|---|---|---|---|
| Pangu-7B | 87.87 | 3,665 | 447.4 | 95.6 | 1.912 |
| *w/o in context examples* | 85.67 | 3,894 | 455.1 | 95.6 | 2.032 |
| Qwen2.5-7B-Instruct | 86.70 | 3,722 | 459.2 | 95.0 | 1.982 |
| *w/o in context examples* | 83.48 | 3,938 | 474.2 | 94.8 | 2.103 |

As shown in Table 4, Pangu-7B outperforms Qwen2.5-7B-Instruct in annotation accuracy (87.87% vs. 86.70%), speeds up thinking trimming (3,665s vs. 3,722s), and reduces total tokens (1.912M vs. 1.982M). Downstream accuracy on MATH500 is essentially unchanged (95.6% vs. 95.0%), showing that occasional consistency errors have negligible effect. Omitting in-context demonstrations in the verifier's prompt (in-context examples shown in Appendix F) slightly increases runtime and token use of the LRMs when the verifiers are applied to the downstream task (Pangu-7B: +229s, +0.120M; Qwen: +216s, +0.121M). Limited by space, the full results using Qwen as verifier are presented in Appendix P. Overall, the choice of verifier model has negligible effects on the overall performance.

**TrimR in Test-time Scaling with BoN.** Beyond sequential token extension, additional test-time scaling approaches to improve LRMs accuracy include BoN sampling (Lightman et al., 2024), Monte Carlo Tree Search (MCTS) (Wu et al., 2025b), and beam search (Lightman et al., 2024).

Integrating TrimR with BoN (N=8), as evidenced in Table 10, Appendix T, demonstrates significant reductions in token consumption (-13.8-16.2%), and runtime duration (up to 23.3%) while maintaining performance parity (-3.3% on AIME24). These results highlight TrimR's broad applicability across diverse test-time scaling frameworks.

## 5 CONCLUSION

This work introduces TrimR, a training-free, verifier-based framework that dynamically trims reasoning in Large Reasoning Models (LRMs) to eliminate redundant thinking. By leveraging a lightweight pre-trained verifier to truncate unnecessary intermediate steps, TrimR significantly improves inference efficiency without compromising accuracy. Empirical results on MATH500, AIME24/25, and GPQA benchmarks demonstrate up to a 70% reduction in runtime across models, particularly in large-batch industrial settings. We also present T4S, TrimR's online deployment system integrated with Ascend NPUs/vLLM, highlighting TrimR's scalability for high-throughput deployments. By balancing computational efficiency and reasoning rigor, TrimR offers a cost-effective solution for real-world LRM applications, advancing the viability of production-level AI reasoning systems.

## 6 REPRODUCIBILITY STATEMENT

The following materials may help reproduce this work.

- Sec. 3 details the proposed method;
- Sec. 3.6 discusses the system design of the proposed method;
- Algorithms 1 and 2 describe the underthinking/overthinking trimming procedures;
- Appendix F offers the full prompts used in verifiers;
- Appendix H describes the experimental setup in detail;
- This study mainly evaluates on open-sourced LRMs (QwQ-32B, DeepSeek-R1-Distill-Qwen-32B, and Pangu-Pro-MoE (Appendix N)) and open-sourced verifier models (Qwen2.5-7B-Instruct and Pangu-7B).

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

## A    USE OF LARGE LANGUAGE MODELS

We employed LLMs for proofreading the manuscript, while ensuring that all substantive points and contributions remain entirely original.

## B    ALGORITHM OF UNDERTHINKING TRIM

We provide the formulation of underthinking trim in Algorithm 2. The underthinking detection depends on the result of overthinking detection. If a sequence can not converge to a solution within the given budget $R_{thres}$ and $N_{thres}$, we use $stop\_tokens$ (Sec. 3.5) to stop further thinking.

---

**Algorithm 2:** Underthinking Trimming

---

**Input:** Input $X$, underthinking threshold $R_{thres}, N_{thres}$
**Output:** Generated output $Y$
$stopped \leftarrow False$;
$y_t \leftarrow p_\Pi(X)$;
**while** $y_t \neq <eos>$ **do** // LRM iterately generates $y_t$ in while loop
    // Check if this sequence is flagged as overthinking
    // $is\_overthinking = False$ indicates that there are no three consecutive answer-bearing chains that agree on the same solution
    $is\_overthinking \leftarrow check\ Algorithm\ 1\ current\ state$;
    $num\_thoughts \leftarrow len(Think\_Seg(y_{\leq t}))$;
    **if** $y_t = </think>$ **then**
        $stopped = True, break$
    **end**
    **if** $is\_overthinking = False\ and\ t > R_{thres}\% \cdot M\ and\ num\_thoughts > N_{thres}\ and\ stopped = False$ **then**
        $y_t \leftarrow p_\Pi((X, \mathbf{y}_{<t}, stop\_tokens)), stopped = True, break$;
    **else**
        $y_t \leftarrow p_\Pi((X, \mathbf{y}_{<t}, y_t))$
    **end**
**end**
**return** $(\mathbf{y}_{<t}, y_t)$;

---

## C    MORE ANALYSIS OF THINKING LENGTH AND ACCURACY

The distribution of decoding length occurrence and corresponding accuracy of Deepseek-R1-Distill-Qwen-32B (R1Q-32B) on the MATH500 and AIME24 datasets are available in Fig. 5a, 5b. The correlation of the number of thoughts and generated tokens are available in Fig. 5c, 5d.

The Deepseek-R1-Distill-Qwen-32B model is weaker than QwQ-32B and tends to generate long but wrong responses, which is underthinking. The accuracy of R1Q-32B quickly decreases from above 80%~90% to 0% as visualized in Fig. 5a, 5b. In contrast, QwQ-32B still is able to solve some complex questions as in Fig. 2a, 2b. In addition, the repetition occurs more frequently, resulting in non-stopping 32K tokens. The number of thoughts of R1Q-32B linearly correlates with the number of tokens, similar to QwQ-32B in Fig. 2.

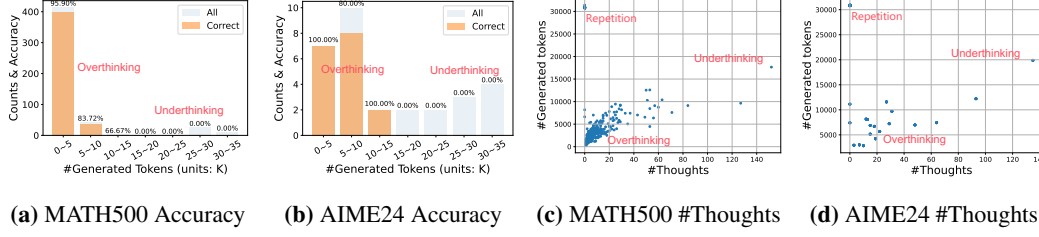

**(a)** MATH500 Accuracy    **(b)** AIME24 Accuracy    **(c)** MATH500 #Thoughts    **(d)** AIME24 #Thoughts

**Figure 5:** Histogram of occurrence and accuracy of decoding length and scatter plot of the number of thoughts and generated tokens of of Deepseek-R1-Distill-Qwen-32B (R1Q-32B) on MATH500 and AIME24 datasets

## D    DECODING THROUGHPUT/LATENCY OVER DECODING LENGTH

Deploying LRMs in large-scale production environments presents substantial challenges for improving reasoning efficiency. First, effective methods to mitigate redundant reasoning in LRMs are

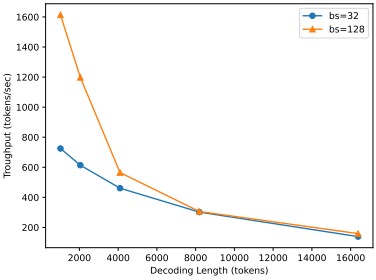

**Figure 6:** Throughput reduction due to increasing length of QwQ-32B. The one-step attention complexity is linear to the sequence length or KV cache length, so inference with long decoding length or long input length has the same issue. For example, for the 32K decoding step, the latency of attention is same as the first decoding step with 32K input length, which is significantly higher than the attention latency with shorter length.

critical, as such inefficiencies significantly hinder the performance of inference systems in production. Since the generation latency of LLMs typically increases linearly with decoding length (as shown in Figure 6), reducing unnecessary token generation can yield super-linear gains in runtime reduction relative to the proportion of tokens saved. This, in turn, enhances the efficiency and scalability of test-time compute. Second, proposed solutions must be compatible with state-of-the-art inference infrastructures designed for large-scale deployment, such as vLLM and SGLang.

## E    FULL LIST OF REFLECTION TOKENS

We utilize the following markers as reflection tokens to partition model reasoning into subthoughts: "\n\nBut", "\n\nWait", "\n\nHowever", "\n\nHmm", "\n\nLet me verify this", and "\n\nAlternatively". We do not utilize those without "\n\n" such as "but" and "But" as reflection tokens to reduce the number of answer existence checking with verifiers, because many are in the internal step checking before approaching answers. "\n\n" is a strong structural separator for different thoughts in Deepseek R1 and Qwen QwQ-32B.

These reflection tokens were chosen by prompting the LRMs with a small set of manually-crafted questions and then capturing the outstanding reflection phrases. Some of the questions are presented as follows:

- Emma walks into her kitchen in the morning and finds the floor wet and several small puddles near the sink. She notices the faucet handle is turned on, but no water is flowing. What most likely happened, and what should Emma do next?

- An A/B test shows a +2.1% uplift with p=0.049 on day 7, but on day 14 the effect shrinks to +0.7% with p=0.18. Provide at least two explanations and the next diagnostic checks.

- A classifier has 98% accuracy on a dataset where 97% of samples are negative. Why might this be misleading? Propose better metrics and a quick verification.

(Continue on the next page)

# F    DETAILED PROMPTS

---

**Default Verifier Prompt for Answer Existence $p_1$**

Check if the following sentences from a reasoning model reach a conclusion, regardless of formatting or phrasing, and solve the question. Return 'Yes' if the content finds a solution, otherwise 'No'. Return only 'Yes' or 'No' with no explanation.
**Example 1:**
**Question:** $2 + 3 =$?
**Content:** The answer is 5.
You should return Yes.
**Example 2:**
**Question:** $2 + 3 =$?
**Content:** I think it should be 5, but I am not sure.
You should return Yes.
**Example 3:**
**Question:** $2 + 3 =$?
**Content:** Wait, I think I made a mistake.
You should return Yes.
**Example 4:**
**Question:** If $f(x) = \frac{3x-2}{x-2}$, what is the value of $f(-2) + f(-1) + f(0)$? Express your answer as a common fraction.
**Content:** 6 + 5 is 11, and 11 + 3 is 14. Yes, so $\frac{14}{3}$. So, $f(-2) + f(-1) + f(0) = \frac{14}{3}$.
You should return Yes.
**Example 5:**
**Question:** If $f(x) = \frac{3x-2}{x-2}$, what is the value of $f(-2) + f(-1) + f(0)$? Express your answer as a common fraction.
**Content:** Since all denominators are 3, we can add the numerators: 6 + 5 + 3 = 14. Therefore, the sum is $\frac{14}{3}$.
You should return Yes.
**Example 6:**
**Question:** If $f(x) = \frac{3x-2}{x-2}$, what is the value of $f(-2) + f(-1) + f(0)$? Express your answer as a common fraction.
**Content:** Wait, another thought: When adding the fractions, is $\frac{14}{3}$ the correct sum? Let's compute it in decimal to cross-verify. $\frac{14}{3}$ divided is approximately 4.666...
You should return No.
**Question:** *$QUESTION*
**Content:** *$CONTENT*
[Yes/No]:

---

**Verifier Prompt for Answer Consistency $p_2$**

Check if the two answers provided are logically or mathematically equivalent, regardless of formatting or phrasing. Return 'Yes' if they are equal in meaning/value and a valid solution to the question, otherwise 'No'. Return only 'Yes' or 'No' with no explanation.

Example 1:
Question: $2 + 3 =$?
Answer1: the answer is 5.
Answer2: the answer seems to be five.
Are they equivalent? [Yes/No]: Yes

Example 2:
Question: Define

$$p = \sum_{k=1}^{\infty} \frac{1}{k^2} \quad \text{and} \quad q = \sum_{k=1}^{\infty} \frac{1}{k^3}.$$

Find a way to write

$$\sum_{j=1}^{\infty} \sum_{k=1}^{\infty} \frac{1}{(j+k)^3}$$

in terms of $p$ and $q$.
Answer 1:
$$(p - 1) - (q - 1) = p - 1 - q + 1 = p - q$$

Therefore, the original double sum is equal to $p - q$.
Answer 2: Given that the terms are decreasing and positive, the convergence seems plausible. However, since $p - q \approx 0.4429$ and our partial sum is about 0.3513, it's possible that the analytical result is correct. Therefore, unless there's a mistake in the analytical steps, the answer should be $p - q$.
Are they equivalent? [Yes/No]: Yes

Question: *$QUESTION*
Answer 1: *$ANSWER1*
Answer 2: *$ANSWER2*
Are they equivalent? [Yes/No]:

---

---

**Guidance Prompt to Stop Thinking**

**Gentle Prompt (Overthinking Trim):**

`\n\n I think I already thought for a long time, and I should be able to put the simple final answer after **Final Answer**\n with \boxed{} and end thinking now!` **\*\*Final Answer\*\*\n**

**Forceful Prompt (Underthinking & Repetition Truncation):**

`\n\n I think I already thought for a long time, and I should be able to deduce the final answer from the content above.` **</think>\n\n**

---

## G  OVERTHINKING/UNDERTHINKING EXAMPLES

The following is an example of overthinking in QwQ-32B. The question comes from AIME24. During the thinking process, reflection tokens like "\n\nWait", "\n\nBut" and "\n\nAlternatively" appear 14 times. After the reflection tokens, the content verifies the answer "204" which first appears in the first third of the entire reasoning process.

---

**An example of overthinking phenomenon (QwQ-32B)**

**Question**: Every morning Aya goes for a 9-kilometer-long walk and stops at a coffee shop afterwards. When she walks at a constant speed of $s$ kilometers per hour, the walk takes her 4 hours, including $t$ minutes spent in the coffee shop. When she walks $s+2$ kilometers per hour, the walk takes her 2 hours and 24 minutes, including $t$ minutes spent in the coffee shop. Suppose Aya walks at $s + \frac{1}{2}$ kilometers per hour. Find the number of minutes the walk takes her, including the $t$ minutes spent in the coffee shop.

**Answer**: Okay, so I have this problem about Aya's morning walk and coffee shop visit. Let me try to parse it step by step ...... Convert that to minutes: 3*60 +24= 180+24=204 minutes. Hmm, but let me confirm again. \n\nWait, in the first scenario ...... So is that the answer? \n\nWait ...... which is 204 minutes. \n\nBut let me double-check ...... so 204 is correct. Hmm. Alternatively, maybe I made a mistake in interpreting the total time? \n\nWait ...... which is 204 minutes. \n\nAlternatively ...... So that's consistent. \n\nAlternatively, maybe there's an error in calculation ...... \n\nWait, hold on, let me re-derive that step ...... Adding 24 minutes gives 204 minutes. So I think that's correct. \n\nAlternatively, maybe the question is in minutes per kilometer or something else? ...... \n\nAlternatively, maybe I need to present the answer in hours converted to minutes? ...... so 204 is the answer. \n\nWait, but let me think again. Let me compute everything step by step once more to be thorough ...... Yes, so that's exactly 180 +24=204. Alright, so I think that's solid ...... \n\nAlternatively, maybe the problem expects the answer in hours ...... \n\nAlternatively, maybe I made a mistake in the quadratic equation ...... \n\nAlternatively, maybe I should check with another approach ...... \n\nTherefore, I think 204 is the correct answer. Maybe I was overcomplicating, but better safe than sorry. I'll go with 204. **Final Answer** 204 </think>

---

(Continue on the next page)

The example of underthinking in QwQ-32B is present below. The question taken from AIME24 is difficult and the LRM doesn't reach the correct answer. The reasoning contents switch between different approaches frequently and can't provide an answer confidently. In fact, the figure 37/128 is made up without reasoning and the answer "475" has been checked to be wrong in previous thinking.

---

**An example of underthinking phenomenon (QwQ-32B)**

**Question**: Each vertex of a regular octagon is independently colored either red or blue with equal probability. The probability that the octagon can then be rotated so that all of the blue vertices end up at positions where there were originally red vertices is $\frac{m}{n}$, where $m$ and $n$ are relatively prime positive integers. What is $m + n$?

**Answer**: Okay, so I need to find the probability that ...... \n\nAlternatively, using Burnside's lemma ...... \n\nWait, Burnside's lemma counts the number of orbits, but I'm not sure. \n\nAlternatively, let me think of it as follows ...... \n\nTherefore, this is the same as before. Perhaps the expected value approach ...... \n\nBut expectation of the union is tricky, but maybe we can approximate it ...... \n\nAlternatively, since B is randomly chosen, the events for different elements are not independent, but perhaps we can compute it ...... \n\nAlternatively, since the problem is from an Olympiad, the answer might be 37/128 or something ...... \n\nAlternatively, maybe the correct answer is 37/128? \n\nWait, let me think differently ...... \n\nAlternatively, perhaps the number is equal to the sum over all non-zero rotations of ...... \n\nBut without knowing the overlaps, it's hard ...... This is getting too involved. Given that this is an Olympiad problem, perhaps there's a smarter way ...... \n\nBut without the second term, we can't compute it. \n\nAlternatively, perhaps assume that the events are independent ...... \n\nWait, this is confusing. \n\nAlternatively, if the number of subsets ...... suggests that this count is too low. \n\nAlternatively, perhaps the correct answer is 37/128? \n\nWait ...... \n\nBut the problem says "can be rotated so that all blue vertices end up at originally red vertices" ...... \n\nTherefore, perhaps the count is indeed 219, giving 475. Given that this is a competition problem, and 475 is a plausible answer (as 219+256=475), I think this is the intended answer. \n\nTherefore, I will go with 475. **Final Answer** 475 </think>

---

## H    DETAILED EXPERIMENT SETUP

Benchmark description: **MATH500** (Hendrycks et al., 2021) comprises 500 problems spanning arithmetic, algebra, geometry, and calculus with varying difficulty levels. It evaluates models' ability in complex mathematical formalism, equation solving, and reasoning. **AIME24** (AIME2024) consists of Olympiad-style problems assessing logical deduction and advanced problem-solving skills. **AIME25** (AIME2025) features updated problems from the same competition as AIME24. **GPQA Diamond** (Rein et al., 2024) is a challenging dataset containing 198 multiple-choice questions, written by domain experts in biology, physics, and chemistry.

All the experiments are conducted on servers with 8 Ascend 910B-64GB NPUs and 192-cores Kunpeng-920 CPUs with 1.5TB memory. During decoding, TorchAir (Torch Ascend Intermediate Representation) captures the computation graph to accelerate kernel dispatch and alleviate host-bound bottlenecks. The maximum number of concurrent decoding batches is set to 128. We configure vLLM to pre-allocate eight scheduler steps, thereby reducing scheduling overhead. All dataset requests (e.g., the 500 questions in MATH500) are submitted to vLLM concurrently, and we record key metrics such as total wall-clock time and per-request latency. We employ the Qwen2.5 math evaluation tool to score the solutions (Yang et al., 2024) and apply postprocessing to ensure that formatting quirks (e.g., spacing, notation style) don't penalize valid solutions.

In the BoN experiments, we use Pangu-ORM (close-source) as the Outcome Reward Model to select the best solution from the generated N solutions. The ratio of LRM and ORM is 1:1 in our experiments, although in production this ratio can be much higher.

## I    EFFECTIVENESS OF REPETITION TRUNCATION

As shown in Table 5, applying repetition truncation to DeepSeek-R1-Distill-Qwen-32B yields substantial efficiency gains without sacrificing—and even slightly improving—accuracy: Enabling truncation reduces total runtime by 22.0%, cuts TPR by 50.10%, and decreases token consumption by

**Table 5:** Effects of Repetition Truncation over five GPQA Diamond runs.

| Model | Runtime (H:M:S) | TPR | Accuracy | #Tokens | Detected Repetitions |
|---|---|---|---|---|---|
| R1Q-32B | 4:02:51 | 5355.66 | 0.444 | 3.09M | — |
| *with repetition truncation* | 3:09:26 | 3568.02 | 0.454 | 2.45M | 29 out of 198 |

20.7%, while delivering a 1% accuracy gain. Further analysis of truncated outputs confirms that most early termination occur during later stages of reasoning, where the model becomes stuck in particular attention patterns and fails to generate diverse contents. We also found that the guidance prompt effectively mitigates infinite repetitive outputs by steering the language model toward contextually relevant generation grounded in prior analysis. This module thus serves as an effective, low-overhead component of the T4S, streamlining inference and enhancing answer clarity.

## J UNDERTHINKING THRESHOLDS

We chose our initial configurations for $R_{thres}$ and $N_{thres}$ by prompting the model with a small suite of several challenging calculus problems that are not within the test sets of the benchmark datasets. We inspected model traces: on these problems, Pangu-R-38B typically exhibited underthinking after $\sim$15K tokens, whereas successful solutions consistently converged within 10K tokens and 20 rounds. Guided by these observations, we fixed a uniform $R_{thres} = 50\%$ and $N_{thres} = 20$ for all benchmarks and models, under which performance remained strong.

One of the questions that can easily trigger underthinking is:

$$S_N = \sum_{k=1}^{N} \ln\left(1 + \frac{k}{N}\right) - N\ln 2 + \frac{1}{2}\ln N.$$

**Tasks:**

1. Prove that $S_N$ admits a full asymptotic expansion as $N \to \infty$ via the Euler–Maclaurin formula; compute explicitly the first four terms (including the constant term).

2. Identify the Bernoulli-number contributions and state a rigorous remainder bound in terms of $\|f^{(m)}\|_\infty$ on $[0, 1]$.

3. Discuss numerical stability: which truncation order minimizes the *actual* error for moderately large $N$ (e.g., $N \approx 10^3$–$10^5$)?

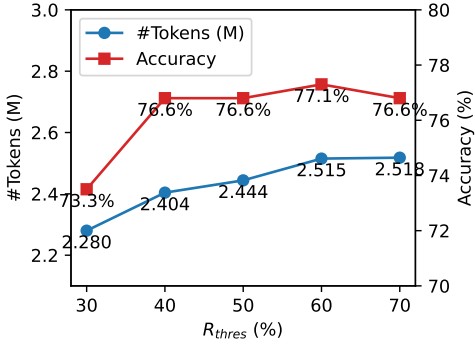

**(a)** Tokens and Acc. vs. $R_{thres}$.

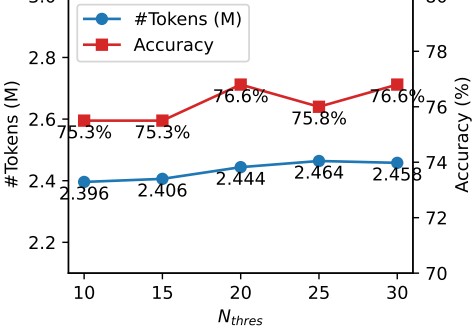

**(b)** Tokens and Acc. vs. $N_{thres}$.

**Figure 7:** Effect of token- and round-budget thresholds on token count and accuracy (QwQ-32B on AIME24). Although performance is largely insensitive to $R_{thres}$ and $N_{thres}$, accuracy degrades when $R_{thres} < 40\%$.

After setting the hyperparameters used for deriving the main results, we change these values and observe the performance as an ablation study. Figure 7 plots accuracy and token usage against the token threshold $R_{thres}$ and round threshold $N_{thres}$. Raising $R_{thres}$ from 30% to 40% increases ac-

curacy from 73.3% to 76.6% while tokens rise modestly (2.28 M to 2.40 M). Beyond 50%, accuracy plateaus but token count continues to grow. Likewise, increasing $N_{\text{thres}}$ from 10 to 20 rounds boosts accuracy to 76.6% (2.44 M tokens) with no clear gains thereafter. Though lowering the threshold to $R_{thres} = 40\%$ yields greater token savings without sacrificing accuracy, our default operating point set by inspection, though not optimal, already offers a reasonably good trade-off between token efficiency and accuracy.

## K    DISCUSSION ON PERFORMANCE CHANGES

We performed a case-by-case review of all instances in which correctness shifted—either positively or negatively—after applying TrimR.

In those cases where accuracy improved, two phenomena were at work. First, on some occasions (approximately 30%), the model would descend into repetitive or lengthy generation and exhaust its token budget without ever producing a final answer; TrimR's early-exit mechanism enables the model to stop and emit its best partial solution instead, turning non-answers into correct outputs. Second, we observed that very long chains of thought can degrade model performance, as global attention becomes less effective over extended sequences. In such samples, the model arrived at a correct solution midway through its reasoning but then later contradicted itself; by detecting convergence and triggering a timely exit, TrimR preserves that correct intermediate answer.

On the other hand, a small fraction of samples saw a slight drop in accuracy. In these cases, the verifier identified a premature convergence on an incorrect hypothesis—often when the model repeatedly reaffirmed the same wrong answer—thereby cutting off subsequent reflections that might have corrected the error. While such instances are uncommon, they highlight a trade-off inherent in any early-exit strategy between minimizing wasted computation and allowing extra time for late-stage corrections. Overall, however, the net effect of TrimR remains strongly positive, yielding significant token savings with only minimal impact on accuracy.

## L    EXTENDED ANALYSIS ON DISTRIBUTION

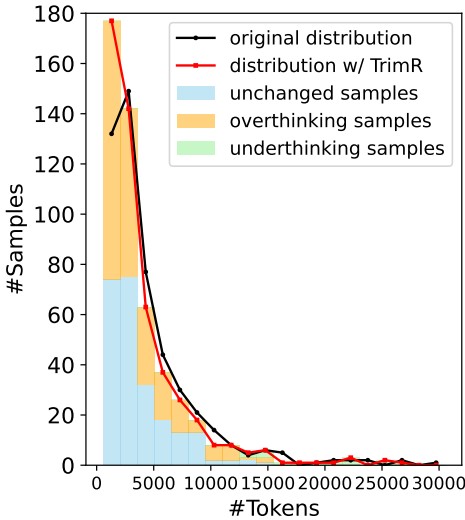
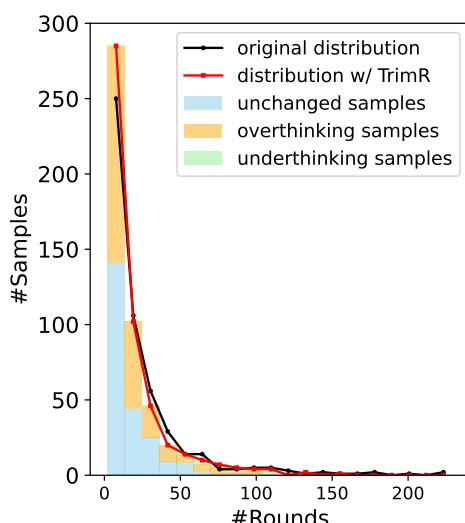

**(a)** The distribution of number of tokens with and without TrimR

**(b)** The distribution of number of rounds with and without TrimR

**Figure 8:** The distribution of tokens and reasoning rounds with and without TrimR (QwQ-32B on MATH500). The original distribution is indicated by the black curve, while the unchanged, overthinking, and underthinking samples are shown as stacked bars. After applying dynamic think trim, both the total tokens and number of reasoning rounds are substantially reduced compared to the original distribution.

As shown in Figure 8a, dynamic thinking trim produces a marked leftward shift in the token–usage distribution. In the uncompressed model, the "knee" lies around 3,500 tokens, with a substantial tail beyond 10,000 tokens. After truncation, over 80% of samples require fewer than 6,000 tokens, a roughly 30% increase. Also, the heavy tails ($>$15,000 tokens) are nearly eliminated. Decomposing the stacked bars reveals that unchanged samples remain tightly clustered in the low-token bins ($\leq$4,000 tokens), overthinking samples are effectively truncated into the lower-range bins (0–8,000 tokens), and underthinking samples ($<$5% of cases) occupy lengths that are modest relative to the original tail.

A parallel effect appears in the reasoning-round distribution (Figure 8b). Prior to trimming, a non-trivial fraction extends beyond 50 rounds (with outliers over 200), whereas after trimming over 85% of samples complete within 30 rounds—an increase of nearly 20 percentage points in the $\leq$20-round regime. Overthinking cases shift from the heavy tail into the 0–30-round interval, while underthinking cases, though rare, have been truncated to mid-range (50–120 rounds).

Taken together, these results show that TrimR preserves valid reasoning, curtails redundant over-thinking, and minimally affects cases needing additional confirmation—thus markedly improving inference efficiency in both token and round dimensions.

## M    ONE VERIFIER CAN SERVE MULTIPLE LRMS

A single verifier model can simultaneously support multiple LRM instances. Runtime performance data indicates that when serving a single LRM instance, the verifier's computational workload remains within manageable thresholds. Each LRM instance generates an average request rate of 9 requests per second, while the verifier demonstrates an average processing capacity of approximately 128 requests per second - establishing an LRM-to-verifier request ratio of 14:1 under ideal conditions. However, production systems adopt a conservative 8:1 deployment ratio to maintain operational safety margins.

When accounting for verifier infrastructure costs, TPR (Time Per Request) improvements must be adjusted by a cost-efficiency factor: $\frac{14 \times 8}{14 \times 8 + 1} = 99.1\%$ (8 Ascend 910-64GB per LRM instance, 1 Ascend 910-64GB per verifier instance). This calculation demonstrates that the performance gains per computational instance remain effectively preserved (99.1%) despite the additional verification overhead. It is noteworthy that enhancements in token utilization and user-perceived TPR cannot be further optimized by scaling the number of computing instances within the cluster. In contrast, TrimR demonstrates significant performance gains by reducing user waiting times, achieving measurable improvements in latency reduction.

## N    PERFORMANCE ON PANGU PRO MOE

We present the results on an open-sourced Pangu model, Pangu Pro MoE, released after the date of this study, in Table 6. The results show that TrimR can readily reduce the token usage with very minor or no performance regression.

**Table 6:** Performance comparison of Pangu–Pro-MoE on the MATH500, AIME24, AIME25, and GPQA Diamond benchmarks. Relative improvements are highlighted in green, and regressions in red.

| Model | Runtime(s) | TPR(s) | TPR-T90(s) | Acc. | #Tokens(M) |
|---|---|---|---|---|---|
| **MATH500** | | | | | |
| Pangu Pro MoE | 6520 | 805.4 | 569.2 | 95.2% | 1.618 |
| *w/ TrimR* | 4900 -24.8% | 662.2 -17.8% | 488.3 -14.2% | 95.6% +0.4% | 1.383 -14.5% |
| **AIME24** | | | | | |
| Pangu Pro MoE | 3047 | 1263.9 | 1096.3 | 80.0% | 0.269 |
| *w/ TrimR* | 2231 -26.8% | 1111.0 -12.1% | 989.1 -9.8% | 76.7% -3.3% | 0.231 -14.1% |
| **AIME25** | | | | | |
| Pangu Pro MoE | 3273 | 1361.7 | 1194.4 | 70.0% | 0.290 |
| *w/ TrimR* | 2256 -31.1% | 1225.8 -10.0% | 1112.3 6.9% | 66.7% -3.3% | 0.256 -11.7% |
| **GPQA Diamond** | | | | | |
| Pangu Pro MoE | 7729 | 1068.0 | 927.3 | 76.6% | 1.347 |
| *w/ TrimR* | 6415 -17.0% | 924.3 -13.5% | 878.8 -15.3% | 76.0% -0.6% | 1.146 -14.9% |

## O  RESULTS ON LIVECODEBENCH

Table 7 shows that, similar to the results on mathematical and scientific reasoning, TrimR can readily reduce token usage by around 10% while preserving the performance of models.

Beyond mathematical and scientific reasoning, the results on LiveCodeBench (Table 7) further demonstrate the robustness of TrimR. TrimR achieves nearly 10% token reduction with accuracy drops consistently below 1%. This trade-off

**Table 7:** Performance and token usage reduction on LiveCodeBench after applying TrimR. Acc.: accuracy change; #Tok.: token usage change.

| Model | Acc. | #Tok. |
|---|---|---|
| QwQ-32B | -0.6% | -9.4% |
| Pangu Pro MoE | -0.8% | -10.2% |

indicates that the method effectively removes redundant exploration while retaining the essential steps needed for correctness. Moreover, its effectiveness across both dense (QwQ-32B) and MoE (Pangu-Pro-MoE) architectures suggests that TrimR is largely architecture-agnostic.

## P  PERFORMANCE OF QWEN 7B AS VERIFIER

The performance of QwQ-32B (LRM) and Qwen2.5-7B-Instruct (verifier) with and without TrimR across benchmarks is presented in Table 8. The results show that the choice of verifier model has negligible effects on TrimR's effectiveness.

**Table 8:** Performance of QwQ-32B (LRM) and Qwen2.5-7B-Instruct (verifier) with and without TrimR across benchmarks. Relative changes are shown in parentheses.

| Dataset | Model | Runtime (s) | TPR (s) | TPR-T90 (s) | Acc. |
|---|---|---|---|---|---|
| MATH500 | QwQ 32B | 4413 | 593.1 | 439.7 | 95.6% |
| MATH500 | QwQ 32B w/ TrimR | 3169 (−28.2%) | 507.1 (−14.5%) | 380.3 (−13.5%) | 96.6% (+1.0%) |
| AIME25 | QwQ 32B | 7436 | 2771.5 | 2513.8 | 60.0% |
| AIME25 | QwQ 32B w/ TrimR | 6321 (−15.0%) | 2342.2 (−15.5%) | 2061.3 (−18.0%) | 60.8% (+0.8%) |
| GPQA Diamond | QwQ 32B | 4406 | 1302.6 | 1115.0 | 66.0% |
| GPQA Diamond | QwQ 32B w/ TrimR | 3305 (−25.0%) | 1185.4 (−9.0%) | 1037.0 (−7.0%) | 66.0% (0%) |

## Q  TIMING OF EARLY STOPPING

We summarize an additional study examining verifier behavior under multi-step checking.

We analyze the trigger timing of early-stop signals by comparing them against human-annotated gold labels. Gold stopping points are defined as three consecutive reasoning segments that lead to semantically equivalent answers. Using both Qwen-based and Pangu-based verifiers, early-stop

signals were triggered on average 1.44 and 1.37 reasoning turns later than the ground-truth optimal stopping point, respectively. Importantly, only 0.5% (Qwen) and 0% (Pangu) of early-stop signals were activated earlier than the human-annotated gold labels.

These observations suggest that although per-step verifier accuracy is not perfect, premature stopping events are extremely rare. In over 99% of cases, early-stop signals do not trigger earlier than the gold stopping points, indicating that TrimR remains robust under imperfect single-step verification. Additional discussion and visualizations will be included in the camera-ready version.

## R   VERIFIER MODEL BIAS AND GENERALIZATION

We further analyze the impact of verifier model characteristics on detection behavior and overall system performance. The study focuses on two aspects: verifier scale and cross-model generalization.

To evaluate the effect of model scale, we experimented with larger verifier models, including Qwen 32B and Pangu 38B. Compared with smaller variants, these larger models produced an approximate 1.5% improvement in detection accuracy. This observation indicates that stronger verifier models can enhance verification reliability, while introducing a predictable increase in computational overhead during the verification stage.

We additionally examine cross-model consistency by comparing multiple 7B-scale verifiers from different model families (Qwen, Pangu, and Llama). Across these variants, detection accuracy remained stable within the 85–90% range. No systematic performance degradation was observed when switching between verifier architectures.

Overall, these results suggest that verifier capability primarily affects the accuracy–efficiency trade-off, while the trimming framework itself exhibits stable behavior across different verifier model families.

## S   CONVERGENCE COUNTER DYNAMICS AND SENSITIVITY TO $M$

We analyze the behavior of the convergence counter used in TrimR and evaluate the impact of alternative counter update rules. In particular, we compare the default reset-based logic against a non-resetting variant. Under the non-resetting strategy, early-stop signals were triggered more frequently, yielding an approximate 4% reduction in token usage. However, this efficiency gain was accompanied by notable accuracy degradation (AIME24/25: $-6.6\%$, MATH500: $-2.1\%$). These results indicate that reset-based counter dynamics provide more reliable convergence behavior, preventing premature termination under transient reasoning fluctuations.

We further conduct an ablation study over the termination threshold parameter $M \in \{1, 2, 3\}$. Table 9 summarizes the relative changes in accuracy and token consumption.

**Table 9:** QwQ-32B — Effect of $M$ on Performance (relative changes in Accuracy / #Tokens)

| Setting | MATH500 | AIME24 | GPQA Diamond |
|---|---|---|---|
| $M = 1$ | $-3\%$ / $-6\%$ | $-6.6\%$ / $-8\%$ | $-2.1\%$ / $-9\%$ |
| $M = 2$ (Baseline) | $0\%$ / $0\%$ | $0\%$ / $0\%$ | $0\%$ / $0\%$ |
| $M = 3$ | $+0.3\%$ / $+8\%$ | $0\%$ / $+7.7\%$ | $+0.2\%$ / $+5.4\%$ |

The results show that $M = 1$ significantly harms accuracy, suggesting insufficient allowance for self-correction. Increasing the threshold to $M = 3$ produces only marginal accuracy improvements while incurring a noticeable efficiency penalty. The default choice $M = 2$ consistently provides the best balance between accuracy and computational efficiency.

## T   TEST-TIME SCALING WITH BoN

Beyond sequential token extension, additional test-time scaling approaches to improve LRMs accuracy include BoN sampling (Lightman et al., 2024), Monte Carlo Tree Search (MCTS) (Wu et al., 2025b), and beam search (Lightman et al., 2024). Integrating TrimR with BoN (N=8), as evidenced

in Table 10, demonstrates significant reductions in token consumption (-13.8-16.2%), and runtime duration (up to 23.3%) while maintaining performance parity (-3.3% on AIME24).

**Table 10:** Integrating TrimR with Best-of-N (BoN) (Pangu-R-38B) yields comparable efficiency improvements and token reduction while preserving the accuracy-performance trade-offs.

| | MATH500 | | | | AIME24 | | | |
|---|---|---|---|---|---|---|---|---|
| | Runtime | TPS | Acc.(%) | #Tokens(M) | Runtime | TPS | Acc.(%) | #Tokens(M) |
| BoN | 19,423 | 656 | 97.4 | 13.055 | 35,084 | 1,628 | 86.6 | 18.882 |
| w/ TrimR | 16,171 -16.7% | 562 -14.3% | 97.2 -0.2 | 11.251 -13.8% | 26,911 -23.3% | 1,281 -21.3% | 83.3 -3.3% | 15.822 -16.2% |

## U  LIMITATIONS

In oder to improve the overthinking and underthinking detection accuracy, we introduce an additional small 7B verifier model. It may require additional hardware for verifiers during deployment, which may increase the complexity of the inference system. However, our algorithm and online system design can dramatically reduce the LRM inference cost. In addition, one verifier can serve multiple LRMs as discussed in Sec. M, so the additional deployment complexity for small verifiers is acceptable compared with the runtime reduction of LRMs. In addition, verifiers can also be deployed in the same hardware with LRMs and they can be collaborated in the way like serving smaller speculative models and larger generation models in speculative decoding.

TrimR currently relies on a curated set of reflection tokens to perform reasoning segmentation. While such lexical markers are widely adopted in recent LRM designs, future models may exhibit weaker or entirely different structural cues. Although segmentation can be re-adapted via corpus analysis, sampling-based updates, or learned segmentation modules, the robustness of this component may vary across model paradigms. Empirically, manual annotation of sampled traces suggests high segmentation accuracy under current models; however, adapting to models without explicit reasoning markers remains an open direction for future work.

## V  BOARDER IMPACTS

Large Reasoning Models are widely used to provide more guidance with internal thinking process and improve user experience. Our work improves the efficiency of LRMs, which is helpful to reduce the cost and carbon footprint of LRMs. This approach can enhance the accessibility of LRMs for a broader population while minimizing environmental impact.

