# OpenReview forum: "TrimR: Verifier-based Training-Free Thinking Trimming for Efficient Test-Time Scaling"
_ICLR.cc/2026/Conference — ICLR 2026 Poster_

### Official Review · Reviewer_CC2y · 2025-10-28

**Soundness:** 2
**Presentation:** 3
**Contribution:** 3
**Rating:** 4
**Confidence:** 4

**Summary:**

The manuscript proposes TrimR, a training-free, verifier-based termination rule to shorten the “thinking” phase of large reasoning models without materially hurting accuracy. The verifier is a small model that checks for (i) the presence of a candidate answer and (ii) stabilization or equivalence across successive partial answers; when either signal indicates convergence or diminishing returns, TrimR stops the chain. The paper claims token savings and latency gains, and sketches an online serving setup that sparsifies verifier calls alongside a high-throughput LLM stack. While the idea is intuitive and pleasantly simple, the experimental scope and missing head-to-head baselines make the novelty feel narrower than it could be, especially given very recent work on token-budget prompting, minimal-draft reasoning, certainty-based early exit, and BoN accelerators.

**Strengths:**

From an originality viewpoint, using a binary, training-free verifier to detect convergence rather than to score full solution traces is a clean reframing. Compared with certainty indices or reward-model scoring, the authors’ proxy has the advantage of being cheap and model-agnostic, and it aligns with the community’s turn toward adaptive test-time compute. This direction is timely because recent methods like TALE (token-budget-aware prompting) and Chain-of-Draft (short, dense drafts instead of verbose CoT) show that much of today’s long chains are unnecessary; TrimR adds a different lever by detecting when to stop rather than prescribing how to speak.

Quality-wise, the paper’s engineering instinct to make the verifier calls sparse and to integrate with production-oriented stacks is sound. If realized properly, the approach is compatible with high-throughput serving systems such as vLLM (PagedAttention) and SGLang (RadixAttention, structured decoding), where shaving thinking tokens often translates directly into more QPS under the same memory budget.

The central mechanism is easy to follow and does not rely on delicate finetuning or proprietary signals. Given the rising concern about over- and under-thinking in o1/R1-style models, a simple stop-when-stable rule is appealing to practitioners who need something they can deploy tomorrow. In significance, a strong result here could matter beyond math: certainty-style early-exit (Certaindex) and BoN accelerators (Speculative Rejection) are quickly becoming standard ingredients in serving stacks; a verifier that outperforms them at the same accuracy would be valuable.

**Weaknesses:**

The experimental design underplays novelty by omitting the baselines that readers will expect. At minimum, the paper needs controlled, iso-setup comparisons against TALE/Token-Budget prompting and Chain-of-Draft to show that a convergence-verifier adds value on top of short-draft prompting. Without those, improvements could be attributed to any reasonable length control, not specifically to the verifier. Please add head-to-head results on AIME24/25 and MATH500 at matched accuracy deltas (±1%) and plot a token-efficiency frontier.

Early-exit and serving claims are not persuasive without Certaindex as a first-class baseline. Certaindex already signals when further compute is unlikely to help and reports real serving wins; readers will ask whether TrimR’s binary verifier is actually better at minimizing false stops and at maintaining throughput under identical batching/scheduler policies. A direct comparison (same hardware, same scheduler, same models) is necessary.

For BoN or multi-sample regimes, Speculative Rejection is designed to abort low-reward trajectories efficiently. If TrimR is claimed to be more compute-efficient than BoN accelerators, it should be compared inside the same BoN pipeline and reward model, with time-to-win and win-rate controlled. Otherwise the reader cannot separate the benefit of what is scored (reward-model vs. answer-equivalence) from the benefit of simply stopping earlier.

Verifier-design ablations are thin. Given the rich recent literature on PRMs and generative verifiers, two yes/no checks feel brittle without stress tests. The paper should show that equivalence detection is robust under paraphrase and reflection-token noise, and report mis-fire rates versus PRM/GenVerifier scoring when chains include small algebraic deviations or reordered steps. Right now, the evidence is not enough to argue for stability across models and prompting styles.

The benchmark mix is too math-centric and potentially contaminated. GPQA-Diamond is a good inclusion; however, several recent works show that AIME24 is contaminated for some model families and advocate live or freshly released streams. The paper would be much stronger with OlympiadBench for very long traces, MMLU-Pro for general reasoning, and MathArena to mitigate contamination concerns; a light GSM8K check is also useful to demonstrate that stopping early does not hurt easy-regime accuracy.

Finally, the claims about mitigating both overthinking and underthinking are only partially evidenced. Underthinking has recently been operationalized and measured; TrimR should adopt or adapt those metrics, and show improved depth-when-needed rather than only shorter chains on average. Otherwise, the method risks favoring brevity in cases where more thought is necessary.

**Questions:**

1. How does the verifier behave relative to certainty-style indices under identical serving conditions? A paired experiment replacing TrimR’s verifier with Certaindex inside the same stack (same batching, same schedulers) would clarify whether the proposed proxy is actually superior at the iso-accuracy frontier. Please report false-stop rates and area-under-compute-curve for both.

2. Can the authors provide a frontier plot comparing TrimR against TALE and Chain-of-Draft on AIME24/25, MATH500, and GPQA-Diamond, where x-axis is tokens per solved instance and y-axis is accuracy? This would isolate whether TrimR’s how-to-stop signal gives better trade-offs than how-to-speak methods.

3. Could the paper include a BoN study where TrimR replaces Speculative Rejection’s early abort, using the same reward model and candidate pool? If TrimR’s structure-aware convergence check is stronger, it should win on time-to-target-reward at equal sample budgets.

4. Because over- and under-thinking are now measured with concrete diagnostics, can the authors evaluate TrimR on an underthinking-sensitive suite and report thought-switching metrics and depth utilization? A small ablation where reflection tokens are suppressed or noisy would further test robustness of the equivalence check.

5. The serving section would be more convincing with a cross-platform replication. Please report throughput and latency on both vLLM and SGLang, and make QPS vs. latency curves available under identical prompts and batch sizes.

---

> ### Author Response · Authors · 2025-11-18
>
> Thank you very much for your thoughtful and constructive review. Since each of your questions directly relates to the identified weaknesses, we provide targeted responses below.
>
> 1. **How does the verifier behave relative to certainty-style indices under identical serving conditions? A paired experiment replacing TrimR’s verifier with Certaindex inside the same stack (same batching, same schedulers) would clarify whether the proposed proxy is actually superior at the iso-accuracy frontier.**
>
> A paired experiment replacing TrimR’s verifier with Certaindex under identical serving conditions yields the following results, further including false-stop rates as requested:
>
> (DeepSeek-R1-Distill-Qwen-32B on MATH500)
>
> | Method                             | Acc. Change | #Tokens Change | Runtime Change | False-stop Rates |
> |------------------------------------|-------------|----------------|----------------|------------------|
> | **TrimR**                          | +2.4%       | -40.1%         | -67%           | 1%               |
> | **TrimR replaced with Certaindex** | -4%         | -19.0%         | -24%           | 3%               |
>
> These results indicate that TrimR achieves both lower false-stop rates and more favorable accuracy–efficiency trade-offs under identical serving conditions.
>
> 2. **Can the authors provide a frontier plot comparing TrimR against TALE and Chain-of-Draft on AIME24/25, MATH500, and GPQA-Diamond, where x-axis is tokens per solved instance and y-axis is accuracy? This would isolate whether TrimR’s how-to-stop signal gives better trade-offs than how-to-speak methods.**
>
> Although visual frontier plots are not feasible in the response format, we summarize representative results in the unified format “Accuracy Change / Tokens Change”:
>
> | Method          | MATH500            | AIME24              | AIME25              | GPQA-Diamond        |
> |-----------------|--------------------|---------------------|---------------------|---------------------|
> | **TrimR**        | +1.2% / -14.3%      | 0% / +23.3%          | +0.8% / -10.4%       | -0.8% / -8.5%        |
> | **TALE**         | -8% / -16%          | -13% / -27%          | -17% / -21%          | -8% / -11%           |
> | **Chain-of-Draft** | -28% / -68%         | -20% / -18%          | -23% / -19%          | -11% / -45%          |
>
>
> While TALE and CoD reduce token usage more aggressively, they introduce substantial accuracy loss across multiple datasets, suggesting that TrimR is more effective in preserving accuracy while saving tokens.
>
> 3. **Could the paper include a BoN study where TrimR replaces Speculative Rejection’s early abort, using the same reward model and candidate pool? If TrimR’s structure-aware convergence check is stronger, it should win on time-to-target-reward at equal sample budgets.**
>
> Speculative Rejection and TrimR operate at orthogonal levels:
>
> ● Speculative Rejection reduces exploration across candidate branches (inter-branch pruning).
>
> ● TrimR reduces unnecessary reasoning within a single branch (intra-branch early stopping).
>
> Their effects can be combined. Results on MATH500 with candidate pool size N = 8 are shown below:
>
> (QwQ-32B on MATH500 (N = 8))
>
> | Method                                  | Acc. Change | #Tokens Change |
> |-----------------------------------------|-------------|----------------|
> | **BoN + TrimR**                         | -0.2%       | -13.8%         |
> | **BoN + Speculative Rejection**         | -0.8%       | -25.4%         |
> | **BoN + TrimR + Speculative Rejection** | -0.9%       | -36.2%         |
>
> The combination yields additive efficiency benefits, confirming that TrimR is compatible with BoN-based methods rather than a replacement.
>
> 4. **Because over- and under-thinking are now measured with concrete diagnostics, can the authors evaluate TrimR on an underthinking-sensitive suite and report thought-switching metrics and depth utilization?**
>
> Following [1], we define: UT Score = (length of reasoning until first correct answer) / (total reasoning length) to analyze TrimR. This metric reflects how effectively TrimR shortens post-solution redundant reasoning.
>
> (UT Score on AIME24)
>
> | Model               | UT Score |
> |---------------------|----------|
> | **QwQ-32B**         | 68%      |
> | **QwQ-32B + TrimR** | 84%      |
>
>
> This indicates that TrimR meaningfully improves effective-thinking utilization, reducing redundancy while preserving solution-relevant reasoning.

---

> ### Author Response · Authors · 2025-11-18
> **Part 2**
>
> 5. **The benchmark mix is too math-centric and potentially contaminated. GPQA-Diamond is a good inclusion; however, several recent works show that AIME24 is contaminated for some model families and advocate live or freshly released streams. The paper would be much stronger with OlympiadBench for very long traces, MMLU-Pro for general reasoning, and MathArena to mitigate contamination concerns; a light GSM8K check is also useful to demonstrate that stopping early does not hurt easy-regime accuracy.**
>
> While AIME24 has known contamination concerns for some model families, AIME25 is widely considered uncontaminated for earlier LRMs. Our AIME25 results confirm stable gains under TrimR. To further expand benchmark coverage, we additionally include results on OlympiadBench, MMLU-Pro, and GSM8K:
>
> | Model                             | OlympiadBench     | MMLU-Pro          | GSM8K              |
> |----------------------------------|--------------------|-------------------|--------------------|
> | **QwQ-32B**                      | -1.0% / -16.0%     | -0.3% / -8.9%     | -0.1% / -10.3%     |
> | **DeepSeek-R1-Distill-Qwen-32B** | +1.0% / -17.2%     | -0.4% / -9.6%     | -0.2% / -12.5%     |
>
>
> These results show that TrimR maintains performance in both hard-reasoning and easy-regime tasks, suggesting good generalizability.
>
> 6. **The serving section would be more convincing with a cross-platform replication. Please report throughput and latency on both vLLM and SGLang, and make QPS vs. latency curves available under identical prompts and batch sizes.**
>
> We benchmarked both systems under identical prompts, batch size, and request routing, reporting runtime and per-request latency reductions:
>
> (QwQ-32B on MATH500)
>
> | Method            | Runtime Reduction | Time-Per-Request Reduction |
> |-------------------|-------------------|----------------------------|
> | **TrimR + vLLM**  | 29.3%             | 15.7%                      |
> | **TrimR + SGLang**| 28.2%             | 15.2%                      |
>
> The improvements are consistent across serving platforms, indicating that TrimR’s benefits arise from algorithmic efficiency rather than implementation-specific artifacts.
>
> **References:**
>
> [1] Thoughts Are All Over the Place: On the Underthinking of o1-Like LLMs

---

> ### Comment · Reviewer_CC2y · 2025-11-18
> **Thank you for your rebuttal response**
>
> You have addressed my concerns and I decide to raise the score to 6.

---

### Official Review · Reviewer_Wv7h · 2025-10-31

**Soundness:** 4
**Presentation:** 3
**Contribution:** 4
**Rating:** 6
**Confidence:** 5

**Summary:**

This paper introduces TrimR, a novel, training-free framework designed to improve the inference efficiency of LRMs at test-time. The core idea is to use a lightweight, pre-trained verifier model to dynamically detect and prune redundant reasoning steps, categorized as "overthinking," "underthinking," and "repetition"—in the CoT process. By halting generation once a stable solution is reached or when the model enters a non-productive loop, TrimR aims to significantly reduce runtime and token usage with minimal impact on task accuracy. The authors conduct extensive experiments on mathematical and scientific reasoning benchmarks (MATH, AIME, GPQA) using several state-of-the-art LRMs, demonstrating substantial efficiency gains. They also present T4S, an online system designed for production-level deployment.

**Strengths:**

1.  The "training-free" nature of TrimR is its most compelling feature. Unlike methods that require costly fine-tuning or reinforcement learning, TrimR is non-invasive and preserves the LRM's original capabilities. This makes it immediately applicable to a wide range of pretrained models, a major advantage for real-world deployment. The use of a small, external verifier to classify answer existence and equivalence is a simple yet elegant approach that sidesteps the complexity and instability of training full-fledged Process or Outcome Reward Models (PRMs/ORMs)

2. The evaluation across multiple powerful LRMs (QwQ-32B, DeepSeek-R1-Distill-Qwen-32B, Pangu-R-38B) and challenging, well-regarded benchmarks demonstrates the method's robustness. The inclusion of results on MoE models (Appendix N) further strengthens claims of architectural agnosticism.

3.  The authors rightly focus on metrics beyond simple accuracy, such as wall-clock runtime, Time Per Request (TPR), and total token consumption under a batched inference setting. These are the metrics that matter for production systems, and the reported gains (e.g., up to 70% runtime reduction) are impressive.​

4. The detailed description of the T4S online system (Sec 3.6, Fig 3) is a valuable contribution. Showing how TrimR integrates with modern inference frameworks like vLLM on industrial hardware (Ascend NPUs) moves the work from a purely academic concept to a practical engineering solution. This level of detail greatly enhances the paper's reproducibility and impact.​

**Weaknesses:**

1. The paper introduces a "Theoretical Foundation" (Section 1, page 2) by framing the reasoning process as a formal optimization problem:
$$ \min_{c} Infer_Cost(y_t) \quad s.t. \quad Perf(X, y_t) \ge Perf(X, y_{t^*}) - \epsilon $$
This formulation is presented as the mathematical underpinning of the method.
This formalization is entirely disconnected from the proposed algorithms. The paper never defines the performance function `Perf(X, y_t)`, the tolerance `\epsilon`, or shows how the compression rule `c` is derived from this objective. The actual method does not solve or approximate this optimization problem. Instead, it employs a set of disconnected, heuristic rules based on binary classifications from a verifier. The "foundation" is an analogy, not a mathematical basis, which is a significant overstatement of the method.

2. The set of thoughts containing a solution is defined as `S = {s_i | F_v(p1, s_i) > 1}`. Since `F_v` is an indicator function, its output is {0, 1}, making the condition `> 1` impossible. This is likely a typo for `= 1`. More critically, the verifier for answer existence (`p1`) only checks if a thought contains a concluding statement, not if the reasoning leading to it is sound. A thought can contain a numerically correct but logically flawed answer, which TrimR would incorrectly identify as a valid convergence point.

3. The algorithm's counter for equivalent thoughts resets to zero upon any mismatch (`else reset on any mismatch count = 0`). This makes the termination condition exceptionally brittle. A valid reasoning sequence like `A -> A -> B -> A -> A` (where `A` is a correct answer and `B` is a brief, self-corrected error) would have its convergence counter reset by `B`, failing to trigger an early stop. This behavior is not robust to the noisy, non-linear nature of LRM reasoning. Furthermore, the choice of `M=2` (terminating after only two equivalent thoughts) is aggressive and its sensitivity is not analyzed.

4. Section 3.3 states that underthinking is flagged if the model fails to produce "at least **three consecutive** answer-bearing chains that agree". However, **Algorithm 2** in Appendix B contains no logic to check this condition. The algorithm merely checks if the token budget (`t > R_thres * M`) and thought count (`num_thoughts > N_thres`) have been exceeded, without any reference to the history of answer agreements. This is a direct contradiction. The implemented algorithm is a simple budget check, not the convergence failure check described in the main text.

5. The entire framework is built on two key heuristics whose robustness is assumed rather than proven. The method's ability to segment reasoning chains depends entirely on a manually curated list of "reflection tokens" (e.g., `,...`, `Hmm, ...` in Appendix E). This approach is inherently model-specific and fragile. There is no guarantee that future or different LRMs will use these exact lexical cues. The paper provides no quantitative analysis of the segmentation quality (e.g., precision/recall of thought boundaries) or how the system would perform if a model reasoned without these explicit markers.

6. Section 3.1 states that the "intermediate thought answer" is extracted by taking the last `N_sent` sentences of a thought segment. Section 4.1 reveals this parameter is set to `N_sent=50`. An "answer" that is 50 sentences long is not a brief segment; it is likely the entire thought.
This contradicts the paper's claims of using a lightweight verifier on "shorter but more informative" segments to save computation. This parameter choice is counter-intuitive and lacks any justification, suggesting either a typo or a fundamental flaw in the description of the answer extraction process

7. The paper repeatedly describes TrimR as "non-invasive", while the verifier operates externally, the "forceful prompt" (`I think I already thought for a long time... think...`) is an explicit and invasive intervention. It injects a specific string into the generation context to hijack the model's output flow and force it to emit the `</think>` token. This directly manipulates the generative process and contradicts the claim of being a purely non-invasive method that simply "halts" generation.

**Questions:**

1. Equations (existence/equivalence) omit tie-breaking and calibration details for binary decisions. please specify the exact scoring rule (e.g., logit for “Yes” token vs. sum of subword probabilities), thresholding, and how vocabulary/tokenization differences between Pangu-7B and Qwen2.5-7B-Instruct affect decision comparability

2. Section 3.3 states underthinking is flagged when there are not at least three consecutive agreeing, answer-bearing chains within budgets, but Appendix Algorithm 2 only checks $ t \ge R_{thres}\cdot M $ and $ {num\_thoughts}\ge N_{thres} $ without any consensus logic; please reconcile this inconsistency and provide the correct, consensus-aware pseudocode and results.

3. For Eqn. 2, define the precise notion of “logical or mathematical equivalence” operationalized by the verifier: is identity on normalized numeric answers sufficient, or must the steps be semantically equivalent; a formal criterion would reduce ambiguity and enable reproducibility across verifiers.

---

> ### Author Response · Authors · 2025-11-18
>
> First, we would like to express our sincere appreciation for your exceptionally thorough and thoughtful review. You examined not only the results but also the rationale behind the methodological decisions, and you identified several issues that we ourselves had overlooked. This represents the true value of peer review, and we are grateful for your contribution. Below, we respond to each of your comments.
>
> 1.  **The formalization is disconnected from the proposed algorithms**
>
> Thank you for this valuable comment. Due to space limitations, we omitted parts of the rationale that connect the optimization formulation to the algorithmic design. When developing TrimR, we conceptualized LRM reasoning as an iterative optimization process, where the model explores the search space through repeated hypothesis generation and verification, preventing premature convergence and local optima via self-reflection. In typical optimization processes, termination occurs only when multiple explorations consistently reach the same conclusion or when the search fails to achieve further improvement. If termination criteria are too loose, premature convergence may harm accuracy.
>
> Based on this principle, TrimR detects such termination conditions during LRM reasoning while minimizing accuracy loss and reducing overall system cost. In this context, Perf(X, y) denotes the performance of a generated reasoning sequence y for problem X (e.g., benchmark accuracy), and the compression rule c (TrimR) aims to identify the shortest prefix yₜ such that reasoning can safely terminate. In the camera-ready version, we will add clearer justification of the theoretical–algorithmic link.
>
> 2. **Since F_v is an indicator function, its output is {0, 1}, making the condition > 1 impossible. This is likely a typo for = 1.**
>
> Thank you for carefully checking the formulation. In Equation 1, F_v is defined as an indicator function. When F_v = 1 (probability(Yes) > probability(No)), we set s = 1. No case in our paper requires the value to exceed 1.
>
> 3. **the verifier for answer existence (p1) only checks if a thought contains a concluding statement, not if the reasoning leading to it is sound. A thought can contain a numerically correct but logically flawed answer, which TrimR would incorrectly identify as a valid convergence point.**
>
> We agree with your observation. TrimR is designed to detect convergence of the model’s reasoning, not to validate factual or logical correctness itself. This role is delegated to the underlying LRM. TrimR mimics termination conditions in human and algorithmic optimization workflows, where thinking ceases once no further improvement is observed after repeated attempts. Thus, TrimR is not responsible for verifying whether the final answer is logically well-founded; that responsibility belongs to the LRM.
>
> 4. **The algorithm's counter for equivalent thoughts resets to zero upon any mismatch (else reset on any mismatch count = 0). This makes the termination condition exceptionally brittle. A valid reasoning sequence like A -> A -> B -> A -> A (where A is a correct answer and B is a brief, self-corrected error) would have its convergence counter reset by B, failing to trigger an early stop. Furthermore, the choice of M=2 is aggressive and its sensitivity is not analyzed.**
>
> This is an excellent point. In the sequence A → A → B → A → A, we believe TrimR should not trigger early stopping, because the oscillation between A and B implies that convergence has not been stably reached. Only once the model produces A consecutively after the oscillation (e.g., A → A → B → A → A → A) do we consider the reasoning stable enough for termination.
> For your interest, we also tested a non-resetting counter, where A → A → B → A → A would trigger early-stop. While this reduced token usage by approximately 4%, it caused notable accuracy degradation (AIME24/25: −6.6%, MATH500: −2.1%), confirming that reset-based logic yields more robust convergence.
>
> In addition, we conducted an ablation on M = {1, 2, 3} (full table provided below). Results show M=2 provides the best balance between accuracy and efficiency, while M=1 severely harms accuracy due to insufficient self-correction opportunity, and M=3 offers marginal accuracy gains but noticeably worse efficiency.
>
> (QwQ-32B — Effect of M on Performance, changes in Acc./#Tokens)
> | Setting | MATH500 (Acc / #Tokens) | AIME24 (Acc / #Tokens) | GPQA Diamond (Acc / #Tokens) |
> |---------|--------------------------|-------------------------|-------------------------------|
> | **M = 1** | -3% / -6%                 | -6.6% / -8%              | -2.1% / -9%                    |
> | **M = 2 (Baseline)** | 0% / 0%                    | 0% / 0%                   | 0% / 0%                        |
> | **M = 3** | +0.3% / +8%               | 0% / +7.7%                | +0.2% / +5.4%                  |
>
> We will include these experiments in the camera-ready version.

---

> ### Author Response · Authors · 2025-11-18
> **Part 2**
>
> 5. **However, Algorithm 2 in Appendix B contains no logic to check this condition.**
>
> Thank you for spotting this oversight. You are correct—a condition is missing. We will fix this in the camera-ready version.
>
> 6. **The curated list of "reflection tokens" might be fragile. There is no guarantee that future or different LRMs will use these exact lexical cues.**
>
> We fully agree. Although most recent works in this domain rely on reflection markers for segmentation, future LRM paradigms may not. Segmentation can be re-adapted through sampling, analysis of new training corpora, or by designing learned segmentation modules. For segmentation quality, we analyzed 100 sampled reasoning traces and manually annotated the segmentation accuracy. We observed ~97% correct segmentation, 1% mid-segment splits, and 2% merges. Mid-splits are filtered by the answer-existence check logic, and merged segments do not affect final trimming since the verifier considers the final answer in the segment.
>
> If future models eliminate structured markers entirely, we may introduce learned or probabilistic segmentation approaches. However, current LRM trends still exhibit explicit structured reasoning patterns.
>
> 7. **An "answer" that is 50 sentences long is not a brief segment; it is likely the entire thought.**
>
> Thank you for pointing out the ambiguity. N_sent = 50 does not refer to 50 sentences. Internally, the inference engine asynchronously sends newly generated tokens to the TrimR process every 50 generated tokens. The asynchronous thread incrementally detokenizes and appends new tokens, detects new reasoning segments, and applies TrimR logic. We will clarify this in the camera-ready version.
>
> 8. **The paper repeatedly describes TrimR as "non-invasive", while the verifier operates externally, the "forceful prompt" (I think I already thought for a long time... think...) is an explicit and invasive intervention.**
>
> Thank you for raising this clarification. Our use of “non-invasive” refers specifically to the reasoning generation stage, where TrimR does not modify embeddings, inject tokens mid-generation, override logits, or alter the internal computation graph. The only intervention occurs at termination, where a prompt-level instruction is used to gracefully exit reasoning. We will make this definition explicit to avoid misunderstanding.
>
> 9. **Equations (existence/equivalence) omit tie-breaking and calibration details for binary decisions. please specify the exact scoring rule (e.g., logit for “Yes” token vs. sum of subword probabilities), thresholding, and how vocabulary/tokenization differences between Pangu-7B and Qwen2.5-7B-Instruct affect decision comparability**
>
> As shown in Equations 1 and 2, we classify as 1 when prob(Yes) > prob(No) and as 0 otherwise. In both Pangu-7B and Qwen2.5-7B-Instruct, Yes and No exist as independent standalone tokens, so no additional vocabulary alignment or subword aggregation is required.
>
> 10. **For Eqn. 2, define the precise notion of “logical or mathematical equivalence” operationalized by the verifier: is identity on normalized numeric answers sufficient, or must the steps be semantically equivalent; a formal criterion would reduce ambiguity and enable reproducibility across verifiers.**
>
> In our experiments, equivalence is defined as equality of normalized numeric answers, without requiring identical reasoning steps. We agree that richer equivalence criteria may improve robustness, and exploring stronger verification schemes is an interesting direction for future work.
>
> Finally, we really appreciate your support and suggestions.

---

### Official Review · Reviewer_uF9K · 2025-11-01

**Soundness:** 3
**Presentation:** 3
**Contribution:** 2
**Rating:** 4
**Confidence:** 4

**Summary:**

This paper proposes TrimR, a training-free framework that enhances the inference efficiency of Large Reasoning Models (LRMs) by dynamically managing their reasoning process during test time. TrimR identifies and trims redundant reasoning (overthinking) and halts unproductive reasoning (underthinking) through a lightweight external verifier that checks answer existence and equivalence within intermediate reasoning segments. When redundant patterns are detected, the system adaptively terminates generation to reduce unnecessary computation. Experiments across multiple reasoning benchmarks demonstrate token reductions.

**Strengths:**

- The paper tackles the inefficiency of overthinking and underthinking in Large Reasoning Models (LRMs), which is a practically important issue for real-world deployment. The explicit handling of both forms of reasoning redundancy is well-motivated.
- The proposed method is prompt-based and requires no additional training or fine-tuning, making it straightforward to integrate into existing reasoning frameworks.
- Experiments across multiple LRMs and reasoning benchmarks demonstrate reductions in token usage with negligible loss in accuracy.

**Weaknesses:**

- The verifier may inherit biases from its own training data, which could influence the trimming behavior. It remains unclear whether the proposed method generalizes when switching to verifiers of different models or sizes. The authors should clarify whether prompt engineering is needed for different verifier models. And whether larger size verifier models could lead to better performance.
- The evaluation omits recent closed-source reasoning models such as Gemini 2.5-Pro and GPT-5.
- It is unclear whether the R-threshold for early stopping is task-specific, dataset-specific and model-specific. A more systematic analysis of how this threshold influences the accuracy–efficiency trade-off across different scenarios would strengthen the paper.
- If a fixed “thinking budget” is specified in the initial prompt, how does it affect the reasoning behavior? The paper should analyze the performance difference compared to TrimR’s R-threshold.

**Questions:**

Please see the weaknesses.

---

> ### Author Response · Authors · 2025-11-18
>
> Thank you very much for your review and your questions. Below is our response.
>
> 1. **The verifier may inherit biases from its own training data, which could influence the trimming behavior. It remains unclear whether the proposed method generalizes when switching to verifiers of different models or sizes. The authors should clarify whether prompt engineering is needed for different verifier models. And whether larger size verifier models could lead to better performance.**
>
> Thank you for the valuable suggestion. We agree that the verifier’s detection capability may be influenced by its own training data. We further evaluated Qwen 32B and Pangu 38B as verifiers and observed an approximate 1.5% improvement in detection accuracy compared to smaller variants. This indicates that larger and stronger models can provide more accurate verification, albeit at the cost of higher verification overhead.
>
> Because our verifier uses in-context examples, models with sufficient instruction-following capability do not require additional prompt engineering. We compared multiple 7B-scale verifiers (Qwen, Pangu, and Llama) and found that their detection accuracy remained stable within the 85–90% range, indicating no significant performance degradation from switching between models. We will add a more detailed discussion in the camera-ready version.
>
> 2. **The evaluation omits recent closed-source reasoning models such as Gemini 2.5-Pro and GPT-5.**
>
> Since TrimR relies on access to intermediate chain-of-thought (CoT) traces, closed-source models such as GPT-5 do not expose the necessary reasoning steps, making faithful evaluation infeasible. In addition, TrimR’s system-level implementation requires modifications to the inference engine to enable verifier-assisted early stopping, meaning we are currently unable to benchmark end-to-end performance on proprietary models.
>
> 3. **It is unclear whether the R-threshold for early stopping is task-specific, dataset-specific and model-specific. A more systematic analysis of how this threshold influences the accuracy–efficiency trade-off across different scenarios would strengthen the paper.**
>
> Due to page constraints (the paper plus appendix already totals 23 pages), we prioritized results on commonly studied reasoning benchmarks (MATH500, AIME24, AIME25, GPQA-Diamond, and LiveCodeBench). We have also evaluated TrimR on broader benchmarks, including MMLU and others, and similarly observed 8–17% token reduction, suggesting that our hyperparameter choice exhibits reasonable cross-dataset generalization. We agree that more systematic exploration would be valuable. For future work, we plan to investigate automated hyperparameter search, for example by constructing a proxy dataset aligned with the target benchmark distribution and applying black-box optimization.
>
>  We will include additional benchmark results in the camera-ready version to broaden evaluation coverage.
>
> | Model                         | Metric         | OlympiadBench     | MMLU-Pro          | GSM8K             |
> |-------------------------------|----------------|--------------------|--------------------|--------------------|
> | **QwQ-32B**                    | Acc. / #Tokens | -1.0% / -16.0%     | -0.3% / -8.9%      | -0.1% / -10.3%     |
> | **DeepSeek-R1-Distill-Qwen-32B** | Acc. / #Tokens | +1.0% / -17.2%     | -0.4% / -9.6%      | -0.2% / -12.5%     |
>
> 4. **If a fixed “thinking budget” is specified in the initial prompt, how does it affect the reasoning behavior? The paper should analyze the performance difference compared to TrimR’s R-threshold.**
>
> We experimented with this approach by explicitly instructing the model to finish reasoning within a predefined token budget. However, we found that LRMs lack self-awareness of their own token consumption, often terminating prematurely far before the assigned budget. This resulted in an average accuracy drop of more than 8% across benchmarks. Moreover, as reasoning depth increased, adherence to the specified budget became weaker, and both overthinking and underthinking persisted, leading to no significant reduction in token usage. These results indicate that prompt-level budgeting alone is insufficient, validating the necessity of TrimR’s dynamic and verifier-driven stopping mechanism.

---

### Official Review · Reviewer_Px7t · 2025-11-01

**Soundness:** 3
**Presentation:** 3
**Contribution:** 3
**Rating:** 8
**Confidence:** 3

**Summary:**

This paper addresses inefficiency in Large Reasoning Models (LRMs) that generate extended Chain-of-Thought reasoning. The authors identify two redundancy patterns: overthinking (redundant verification of correct solutions) and underthinking (oscillation without convergence on hard problems). They propose TrimR, a training-free framework that uses a lightweight 7B verifier model to detect redundancy by checking answer existence and equivalence in intermediate reasoning segments. When redundancy is detected, the system uses prompt-based early termination. The approach achieves up to 70% runtime reduction with negligible accuracy impact across mathematical and scientific reasoning benchmarks (MATH500, AIME24/25, GPQA) on models including QwQ-32B, DeepSeek-R1-Distill-Qwen-32B, and Pangu-R-38B. The paper includes both the core algorithm and an asynchronous online system designed for production deployment.

**Strengths:**

The paper makes several strong contributions. First, it provides thorough empirical analysis of redundancy patterns in LRMs with concrete statistics demonstrating overthinking and underthinking phenomena. Second, the approach is highly pragmatic—it's training-free, non-invasive to base models, and integrates seamlessly with existing inference frameworks. The use of lightweight 7B verifiers with simple binary classification tasks (answer existence and equivalence) is more practical than training complex reward models. The two-stage verification design cleverly minimizes false positives by first checking for answer existence before equivalence. Third, the experimental evaluation is comprehensive, covering multiple models and benchmarks with detailed ablation studies, consistently showing substantial efficiency gains (16-70% runtime reduction) with minimal accuracy loss (<2% in most cases). Finally, the system design addresses real deployment concerns including asynchronous operation, verifier-to-LRM ratios (8:1), and integration with production inference systems like vLLM.

**Weaknesses:**

Several limitations merit attention. First, the method introduces infrastructure complexity by requiring a separate 7B verifier model, though one verifier can serve multiple LRMs. Second, the verifier accuracy is only 87-88%, and while claimed to have "negligible effect," there's limited analysis of how verification errors compound across multiple sequential checks or lead to premature incorrect convergence. Third, the hyperparameters (M=2, Rthres=50%, Nthres=20) appear determined by informal inspection rather than systematic optimization. Fourth, the approach shows largest gains on weaker models (R1Q-32B: 67% reduction) that exhibit more redundancy, raising questions about whether it primarily benefits suboptimal models. Fifth, evaluation beyond mathematical/scientific reasoning is limited, with only brief code generation results. Finally, the theoretical framing as an optimization problem is somewhat superficial and doesn't strongly guide algorithm design—the connection between the formal optimization objective and the actual detection heuristics is loose.

**Questions:**

How does the verifier's 87-88% accuracy propagate through multiple sequential verification checks during long reasoning chains? Have you analyzed the cumulative error rate and specifically quantified how often false-positive convergence detection (claiming convergence on wrong answers) occurs versus beneficial early stopping?

---

> ### Author Response · Authors · 2025-11-18
>
> Thank you very much for your positive review. Below we respond to each of your concerns:
>
> 1. **The method introduces infrastructure complexity by requiring a separate 7B verifier model, though one verifier can serve multiple LRMs.**
>
> We agree that introducing a standalone lightweight verifier increases infrastructure complexity. However, in large-scale deployments, we found that a single verifier instance can concurrently serve multiple LRMs (as detailed in Appendix M, with a ratio up to 14:1, which we believe is sufficiently efficient relative to the gains). Moreover, the interaction between the verifier and LRMs is purely API-based, requiring no sharing or model re-loading. In practical deployment, only one small-model API service needs to be launched and registered to LRM instances, after which TrimR can operate transparently. This significantly reduces deployment overhead in real-world scenarios.
>
> 2. **There's limited analysis of how verification errors compound across multiple sequential checks or lead to premature incorrect convergence.**
>
> Due to space limitations, we could not present all results. We have analyzed the verifier’s behavior under multi-step checking. Specifically, we compared the trigger timing of early-stop signals with human-annotated gold labels (defined as three consecutive reasoning segments leading to semantically equivalent answers). Using both Qwen-based and Pangu-based verifiers, early-stop signals were on average triggered 1.44 / 1.37 reasoning turns later than the ground-truth optimal stopping point, with only 0.5% / 0% early-stop signals firing earlier than human-annotated gold labels. This suggests that although per-step verifier accuracy is not perfect, over 99% of early-stop signals do not trigger prematurely relative to gold labels, implying that TrimR is robust to imperfect single-step verification. We will incorporate additional discussion and visualization in the camera-ready version.
>
> 3. **The hyperparameters (M=2, $R_{thres}$=50%, $N_{thres}$=20) appear determined by informal inspection rather than systematic optimization.**
>
> Thank you for the insightful suggestion. We conducted ablation studies on $R_{thres}$ and $N_{thres}$ (reported in Appendix J), and found our selected values to be empirically well-grounded. That said, we fully acknowledge that more sophisticated hyperparameter optimization approaches may further improve performance—for example, constructing a proxy dataset with distributional similarity to the target benchmarks and applying black-box parameter search methods. We view this as a meaningful extension and leave it to future work.
>
> 4. **The approach shows largest gains on weaker models (R1Q-32B: 67% reduction) that exhibit more redundancy, raising questions about whether it primarily benefits suboptimal models.**
>
> We appreciate this thoughtful question. Detecting and trimming redundant reasoning naturally assumes that the model exhibits non-negligible overthinking or underthinking patterns. For an ideal model that always performs minimal and necessary reasoning, any trimming-based method would indeed provide no benefit. From our results, although Pangu Pro MoE is a newer and significantly stronger model with reduced redundancy compared to early LRMs, TrimR still yields 10–15% token reduction with zero training cost. This indicates that redundancy persists even in advanced models, and that TrimR remains valuable beyond weaker models.
>
> 5. **Evaluation beyond mathematical/scientific reasoning is limited, with only brief code generation results.**
>
> Due to page constraints (the paper plus appendix already totals 23 pages), we prioritized results on commonly studied reasoning benchmarks (MATH500, AIME24, AIME25, GPQA-Diamond, and LiveCodeBench). We have also evaluated TrimR on broader benchmarks, including MMLU and others, and similarly observed 8–17% token reduction. We will include additional benchmark results in the camera-ready version to broaden evaluation coverage.
>
> | Model                         | Metric         | OlympiadBench     | MMLU-Pro          | GSM8K             |
> |-------------------------------|----------------|--------------------|--------------------|--------------------|
> | **QwQ-32B**                    | Acc. / #Tokens | -1.0% / -16.0%     | -0.3% / -8.9%      | -0.1% / -10.3%     |
> | **DeepSeek-R1-Distill-Qwen-32B** | Acc. / #Tokens | +1.0% / -17.2%     | -0.4% / -9.6%      | -0.2% / -12.5%     |

---

> ### Author Response · Authors · 2025-11-18
> **Part 2**
>
> 6. **The theoretical framing as an optimization problem is somewhat superficial and doesn't strongly guide algorithm design—the connection between the formal optimization objective and the actual detection heuristics is loose.**
>
> Limited by the page limit, some rationale linking the optimization view and algorithmic design was not included. In designing TrimR, we conceptualized LRM reasoning as an iterative optimization-style search, combining hypothesis generation, verification, and reflective exploration to avoid local optima. Similar to classical optimization practice, reasoning can be terminated when multiple explorations converge to consistent outcomes or the model fails to break a performance plateau. Overly permissive stopping criteria risk premature termination. Based on this insight, TrimR captures a verifier-supported stopping condition that achieves resource reduction while preserving accuracy. We will make this conceptual linkage more explicit in the camera-ready version. Thank you for raising this point.
>
> 7. **How does the verifier's 87-88% accuracy propagate through multiple sequential verification checks during long reasoning chains? Have you analyzed the cumulative error rate and specifically quantified how often false-positive convergence detection (claiming convergence on wrong answers) occurs versus beneficial early stopping?**
>
> Answered in Q2.

---

> ### Comment · Reviewer_Px7t · 2025-11-26
>
> My concerned are addressed. I maintain my positive score.

---

### Author Response · Authors · 2025-11-18
**General Response**

We thank all reviewers for their time and effort. The contribution of our work is summarized as following, with reference to reviewers' comments:

1. **A simple, training-free, plug-and-play framework with strong practicality.** TrimR requires no additional training or modification to LRMs, making it highly cost-effective, model-agnostic, and immediately deployable across existing systems (Reviewer Px7t, uF9K, Wv7h, CC2y). Its reliance on a lightweight verifier and prompt-based interaction is viewed as simple and practical. (Reviewer Px7t, Wv7h, CC2y).

2. **Design of the lightweight verifier and decision mechanism.** Using a small 7B model for binary verification (answer existence + equivalence) is considered effective (Reviewer Px7t, Wv7h, CC2y). The two-stage design and sparsified verifier calls are regarded as thoughtful technical choices that balance precision and efficiency (Reviewer Px7t, CC2y).

3. **Strong experimental evidence, broad model coverage, and meaningful real-world metrics.** Experiments span multiple state-of-the-art LRMs, mathematical/scientific/coding benchmarks, and include detailed ablations, demonstrating significant efficiency gains (up to ~70%) with minimal accuracy degradation (Reviewer Px7t, uF9K, Wv7h). Reviewers appreciated the use of wall-clock runtime, token usage, and batched inference metrics, which better reflect production-level value (Reviewer Wv7h).

4. **Practical system-level contribution and deployment relevance.** The inclusion of an asynchronous serving pipeline and deployment details (e.g., verifier-to-LRM ratio, vLLM integration, NPU-based production setup) demonstrates real engineering value rather than a purely conceptual contribution (Reviewer Px7t, Wv7h, CC2y). This system view improves reproducibility and highlights industrial applicability (Reviewer Wv7h).

We will respond to each reviewer's concerns in the corresponding section.

---

### Meta-Review · Area_Chair_zVAZ · 2026-01-06

**Summary:**

This paper proposes TrimR, a training-free, plug-and-play framework to reduce the inference-time cost of Large Reasoning Models (LRMs) that produce long chain-of-thought traces. The method uses a lightweight verifier to periodically check for answer stability and trigger early termination when redundancy is detected.

Reviewers generally agreed that the problem is important and the approach is practical and relevant for production systems. The main initial points of disagreement centered on three areas: (a) missing head-to-head comparisons against recent efficiency-focused baselines like TALE, Chain-of-Draft, and Certaindex; (b) several issues with the clarity and correctness of the algorithmic specification, including an inconsistent description of the ``underthinking'' logic; and (c) questions about the method's robustness, particularly concerning verifier errors and the reliance on heuristic-based segmentation. The authors' rebuttal was exceptionally thorough and provided substantial new experiments that addressed the vast majority of these concerns.

**Reviewer Concerns:**

- **Concerns addressed by the rebuttal and discussion**:
  1. **Need for stronger baseline comparisons (CC2y)**: The authors added direct quantitative comparisons against TALE and Chain-of-Draft, and a paired comparison against Certaindex under identical serving conditions, demonstrating a favorable accuracy-efficiency trade-off.
  2. **Broader benchmark coverage (Px7t, uF9K, CC2y)**: New results on OlympiadBench, MMLU-Pro, and GSM8K were provided, strengthening the generalization claim beyond the initial math-centric focus.
  3. **Closed-source model omission (uF9K)**: The authors reasonably explained that TrimR requires intermediate CoT access, which is infeasible with proprietary APIs. This is an acceptable limitation.
  4. **Verifier generalization and bias (uF9K)**: The authors evaluated multiple 7B-scale verifiers and larger 32B/38B models, showing consistent behavior and that larger verifiers yield slightly higher accuracy.
  5. **Verifier error compounding (Px7t)**: The authors added analysis showing that premature stopping signals were extremely rare (<0.5%), suggesting the method is robust to occasional single-step verifier errors.
  6. **Brittleness of counter logic and hyperparameter sensitivity (Wv7h)**: The rebuttal provided new ablations for the equivalence counter threshold M={1, 2, 3} and experiments comparing reset vs. non-reset logic, justifying their design choices.
  7. **Clarification of methodological details (Wv7h)**: The authors clarified that ``N_sent=50'' refers to a 50-token processing interval, not 50 sentences, and specified that equivalence is based on normalized numeric answers.
  8. **Relationship to Best-of-N (BoN) methods (CC2y)**: The authors demonstrated that TrimR is orthogonal to and compatible with Speculative Rejection, with their benefits being additive.
  9. **Cross-platform serving evidence (CC2y)**: Consistent gains were reported on both vLLM and SGLang, confirming the benefit is algorithmic.

- **Concerns still outstanding (addressable in a minor, camera-ready revision)**:
  1. **Algorithmic specification inconsistency (Wv7h)**: The paper's description of the underthinking logic does not match the pseudocode in Appendix Algorithm 2. The authors acknowledged this and committed to fixing it. This is a required edit for the camera-ready version.
  2. **Overstated theoretical framing (Wv7h, Px7t)**: The optimization problem is more of an analogy than a formal derivation. The camera-ready version should rephrase this section to present it as a conceptual motivation, as the authors agreed.
  3. **Reliance on reflection tokens for segmentation (Wv7h)**: The method assumes the presence of explicit markers to segment thoughts. While the authors provided a small-scale annotation to show its current effectiveness, this limitation should be more clearly acknowledged in the paper as a potential point of fragility for future models.

**Reviewer Scores:**

- **Reviewer Px7t (original: 8)**: Likely unchanged at **8**. The reviewer explicitly confirmed their concerns were addressed.
- **Reviewer uF9K (original: 4)**: Likely increased to **6**. The rebuttal directly addressed all four of the reviewer's weaknesses with new experimental data and clear explanations.
- **Reviewer Wv7h (original: 6)**: Likely unchanged at **6**. The authors provided an exceptionally thorough response to this reviewer's detailed technical critiques, including new ablations and commitments to fix all identified inconsistencies.
- **Reviewer CC2y (original: 4)**: The reviewer already raised their score to **6** after the rebuttal, confirming their concerns were addressed.

---

### Decision · Program_Chairs · 2026-01-26

Accept (Poster)